https://doi.org/10.1038/s41467-019-10649-4　　**OPEN**

# Improving the diagnostic yield of exome-sequencing by predicting gene–phenotype associations using large-scale gene expression analysis

Patrick Deelen [1,2,4], Sipko van Dam[1,4], Johanna C. Herkert [1], Juha M. Karjalainen[1], Harm Brugge[1], Kristin M. Abbott [1], Cleo C. van Diemen[1], Paul A. van der Zwaag[1], Erica H. Gerkes[1], Evelien Zonneveld-Huijssoon [1], Jelkje J. Boer-Bergsma[1], Pytrik Folkertsma[1], Tessa Gillett[1], K. Joeri van der Velde[1,2], Roan Kanninga[1,2], Peter C. van den Akker[1], Sabrina Z. Jan[1], Edgar T. Hoorntje[1,3], Wouter P. te Rijdt [1,3], Yvonne J. Vos[1], Jan D.H. Jongbloed[1], Conny M.A. van Ravenswaaij-Arts[1], Richard Sinke[1], Birgit Sikkema-Raddatz[1], Wilhelmina S. Kerstjens-Frederikse [1], Morris A. Swertz [1,2] & Lude Franke [1]

The diagnostic yield of exome and genome sequencing remains low (8–70%), due to incomplete knowledge on the genes that cause disease. To improve this, we use RNA-seq data from 31,499 samples to predict which genes cause specific disease phenotypes, and develop GeneNetwork Assisted Diagnostic Optimization (GADO). We show that this unbiased method, which does not rely upon specific knowledge on individual genes, is effective in both identifying previously unknown disease gene associations, and flagging genes that have previously been incorrectly implicated in disease. GADO can be run on www.genenetwork.nl by supplying HPO-terms and a list of genes that contain candidate variants. Finally, applying GADO to a cohort of 61 patients for whom exome-sequencing analysis had not resulted in a genetic diagnosis, yields likely causative genes for ten cases.

[1] University of Groningen, University Medical Center Groningen, Department of Genetics, 9700 VB Groningen, The Netherlands. [2] University of Groningen, University Medical Center Groningen, Genomics Coordination Center, 9700 VB Groningen, The Netherlands. [3] Netherlands Heart Institute, 3511 EP Utrecht, The Netherlands. [4] These authors contributed equally: Patrick Deelen, Sipko van Dam. Correspondence and requests for materials should be addressed to L.F. (email: Lude@ludesign.nl)

D iagnostic yield is steadily improving with the increasing use of whole-exome sequencing (WES) and whole-genome sequencing (WGS) to diagnose patients with a suspected genetic disorder[1]. Although many genes have been associated to Mendelian diseases, the diagnostic yield of genome sequencing remains limited, varying from 8 to 70%[2].

Tools exist that can help prioritize candidate genes based on existing knowledge, of which some use human phenotype ontology (HPO) terms[3] to denote the phenotype of a patient. However, these methods are often limited in their ability to identify previously unknown disease-gene associations[4]. For instance, AMELIE prioritizes candidate genes using an automated literature analysis, but cannot pinpoint genes, unknown to cause a certain disease[5]. In contrast, Exomiser can aid in disease-gene discovery by using existing (knock out) annotations for genes or orthologues in other organisms[6]. Also, the tissue specificity of gene expression has been shown to be informative for predicting disease relevance[7]. While each of these methods have proven highly valuable, one challenge remains: for most protein-coding and non-coding genes very little is known, making it also very challenging to infer whether a mutation in those genes cause a specific phenotype.

Another problem is that some genes or variants that have previously been implicated in the prevalence of a specific disease are now reported as either being false positive associations or having limited penetrance[8, 9]. Often these likely false associations are identified because the presumed causative variant alleles turn out to be too common in large populations, such as present in ExAC[10, 11]. Alternatively, the effects of variants in some genes could not be replicated in population-based biobanks[12]. Although only few genes have been definitely refuted in literature, it has been shown that many genes reported in rare disease databases only have limited evidence to link the gene to the disease[13].

Here, we present a method to overcome some of these challenges. By using 31,499 RNA-sequencing (RNA-seq) samples of a wide range of tissues and cell types, we can predict gene functions and disease associations, while not being biased towards existing gene annotations by using a leave-one-out procedure. Using gene co-regulation allows us to accurately predict gene functions and to prioritize candidate disease genes with high accuracy. This is possible because if genes are known to cause a specific disease or disease symptom they often have similar molecular functions or are involved in the same biological process or pathway[14]. When the reported disease associations cannot be predicted this may indicate false positive associations.

We introduce a user-friendly web-based tool called GADO (GeneNetwork Assisted Diagnostic Optimization, available at www.genenetwork.nl) and a command line version (available at https://github.com/molgenis/systemsgenetics/wiki/GADO-Command-line) that can prioritize variants in known *and* unknown genes using HPO-terms to describe a patient's phenotype. GADO ranks variants using HPO terms to describe a patient's phenotype. We validate our prioritization method by testing how well our method predicts disease-causing genes based on HPO-terms described for each of the genes in the OMIM database. Exome sequencing data of patients with a known genetic diagnosis are used to benchmark GADO. Finally, we apply our methodology to previously inconclusive WES data and identify several genes that contain variants that likely explain the phenotype of the respective patients. Thus, we show that our methodology is successful in identifying variants in previously unknown, likely relevant genes explaining the patient's phenotype.

## Results

**Gene prioritization using GADO.** We have developed GADO, a method that can perform gene prioritizations, which uses as input a list of phenotypes (described using HPO terms[15]) that have been observed in a patient. In combination with a list of candidate genes (i.e., genes harboring rare and possibly damaging variants), GADO reports a ranked list of genes with the most likely candidate genes on top (Fig. 1a). These gene prioritizations are based on the predicted involvement of the candidate genes for the specified set of HPO terms. These predictions are made by analyzing public RNA-seq data from 31,499 samples (Fig. 1b), resulting in a gene prioritization Z-score for each HPO term. These predictions are solely based on observed co-regulation of genes annotated to a certain HPO term with other genes. This makes it possible to also prioritize genes that currently lack any biological annotation.

**Public RNA-seq data acquisition and quality control.** To predict functions of genes and HPO term associations, all human RNA-seq samples that were publicly available in the European Nucleotide Archive (accessed June 30, 2016) were downloaded (Supplementary Data 1)[16]. Gene-expression was quantified by using Kallisto[17] and samples for which a limited number of reads are mapped, were removed. A principal component analysis (PCA) on the correlation matrix was used to remove low quality samples and to remove samples that were falsely annotated as RNA-seq but turned out to be DNA-seq. Finally, 31,499 samples were included and gene expression levels for 56,435 genes (of which 22,375 are protein-coding) were quantified.

Although these samples are generated in many different laboratories, we previously observed that, after correcting for technical biases, it is possible to integrate these samples into a single expression dataset[18]. We validated that this is also true for our dataset by visualizing the data using t-Distributed Stochastic Neighbor Embedding (t-SNE). We labeled the samples based on cell-type or tissue and we observed that samples cluster together based on cell-type or tissue origin (Fig. 2). Technical biases, such as whether single-end or paired-end sequencing had been used, did not lead to erroneous clusters, which suggests that this heterogeneous dataset can be used to ascertain co-regulation between genes and can thus serve as the basis for predicting the functions of genes (Methods).

**Prediction of gene HPO associations and gene functions.** To predict HPO term associations and putative gene functions (Fig. 1b), we used a co-regulation method that we had previously developed and applied to public expression microarrays[14]. However, since microarrays only cover a subset of the protein-coding genes ($n = 14,510$), we decided to use public RNA-seq data here instead. This allows for more accurate quantification of lower expressed genes and the expression quantification of many more genes, including a large number of non-protein-coding genes[19]. Our method uses principal component analysis to identify a set of components that describe co-regulation between genes. While some of this co-regulation between genes is determined by pairs of genes that are specifically expressed in certain tissues (i.e., tissue-specific expression), a considerable proportion of this co-regulation reflects pairs of genes that are involved in the same biological pathways.

We applied this prediction methodology[14] to the HPO gene sets and also to Reactome[20], KEGG pathways[21], Gene Ontology (GO) molecular function, GO biological process and GO cellular component[22] gene sets. For 5,088 of the 8,657 gene sets (59%) with at least 10 genes annotated, the gene function predictions had significant predictive power (see Methods). For the 8,657 gene sets with at least 10 genes annotated, the median predictive power, denoted as Area Under the Curve (AUC), ranged between 0.73 (HPO) to 0.87 (Reactome) (Table 1).

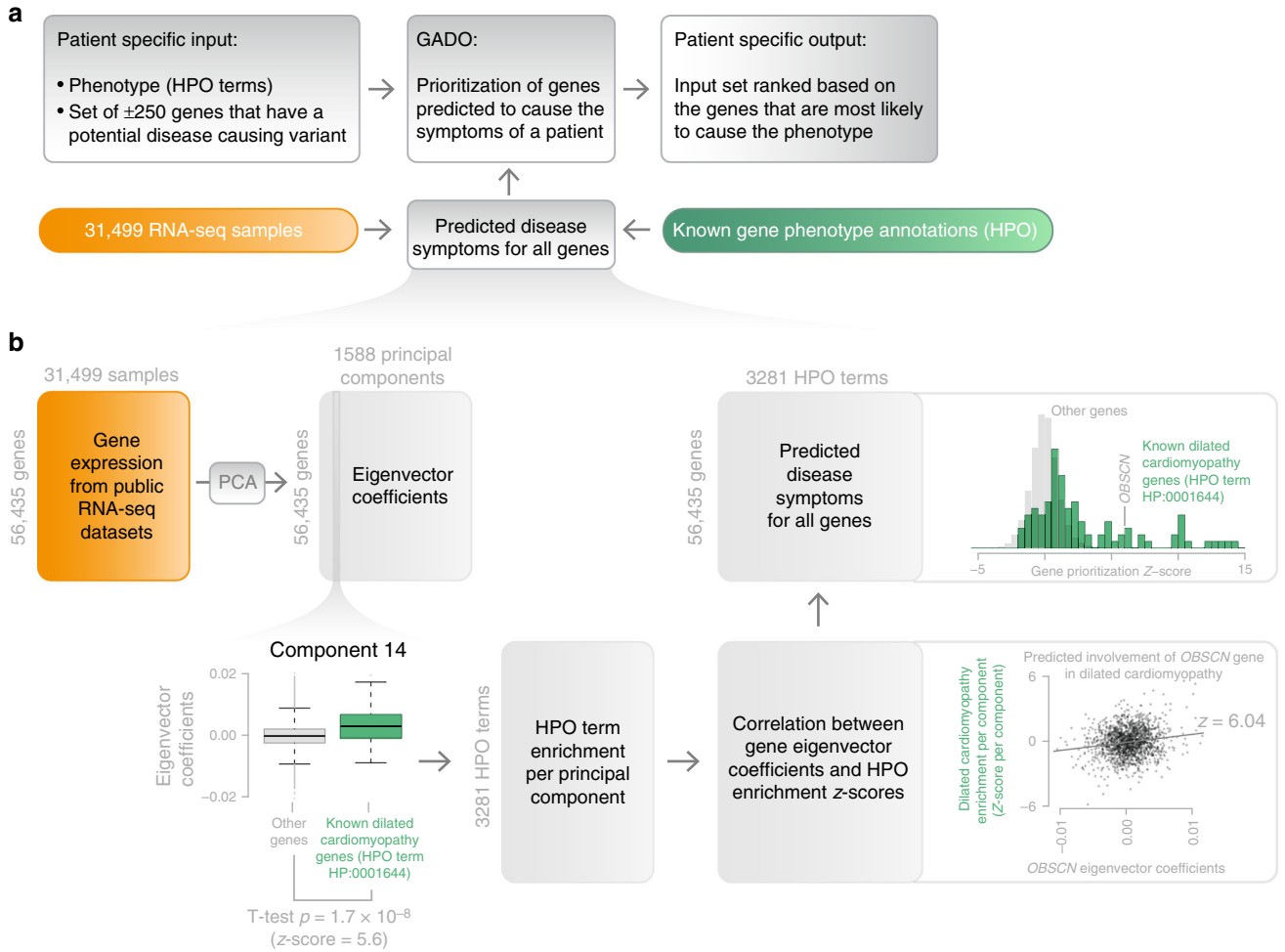

**Fig. 1** Schematic overview of GADO. **a** Per patient, GADO requires a set of phenotypic features (encoded using HPO terms) and a list of candidate genes (gene names either entered using HGNC symbols or Ensembl IDs). This gene list should contain genes in which rare variants have been observed for the patient. It then ascertains whether any of these genes have been predicted to cause the phenotypic features, observed in the patient. These HPO phenotypes predictions per gene are based on observed co-regulation with sets of genes that are already known to be associated with these phenotypes. **b** Overview of how disease symptoms are predicted using gene expression data from 31,499 human RNA-seq samples. A principal component analysis on the co-expression matrix results in the identification of 1588 significant principal components. For each HPO term we investigate every component: per component we test whether there is a significant difference between eigenvector coefficients of genes known to cause a specific phenotype and a background set of genes. This results in a matrix that indicates which principal components are informative for every HPO term. By correlating this matrix to the eigenvector coefficients of every individual gene, it is possible to infer the likely HPO disease phenotype term that would be the result of a pathogenic variant in that gene

**Prioritization of known disease genes using the annotated HPO terms**. Once we had calculated the prioritization Z-scores of HPO disease phenotypes, we leveraged these scores to prioritize genes found by sequencing the DNA of a patient. For each individual HPO term–gene combination, we calculated a prioritization Z-score that can be used to rank genes. In practice, however, patients often present with not one feature but a combination of multiple phenotypic features. Therefore, we combined the prioritization Z-scores for each HPO term to generate an overall prioritization Z-score that explains the full spectrum of features in a patient. GADO uses these combined prioritization Z-scores to prioritize the candidate genes: the higher the combined prioritization Z-score for a gene, the more likely it explains the patient's phenotypes.

Because many HPO terms have fewer than 10 genes annotated, and since we were unable to make significant predictions for some HPO terms, certain HPO terms are not suitable to use for gene prioritization. To overcome this problem we take advantage of the way HPO terms are structured: each term has at least one

parent HPO term that describes a more generic phenotype and thus has also more genes assigned to it. Therefore, if an HPO term cannot be used, GADO will make suggestions for suitable parental terms (Supplementary Fig. 1).

To benchmark our prioritization method, we used the OMIM database[23]. Due to our leave-one-out approach (see methods) we could directly test how well our method was able to retrospectively rank disease-causing genes listed in OMIM based on the annotated symptoms of these diseases. For each OMIM disease gene ($n = 3,382$) we used the associated disease features (on average 15 HPO-terms per gene) as input for GADO. We found that GADO ranks the causative gene in the top 5% for 49% of the diseases (Fig. 3a, Supplementary Fig. 2). However, in clinical practice it is not uncommon that only a subset of the features of a patient have been recorded. We therefore repeated this analysis while randomly selecting at most 5 HPO terms per disease. We found that the GADO scores remained stable and are strongly correlated (Pearson correlation $r = 0.86$) compared to using all HPO terms (Supplementary Fig. 3).

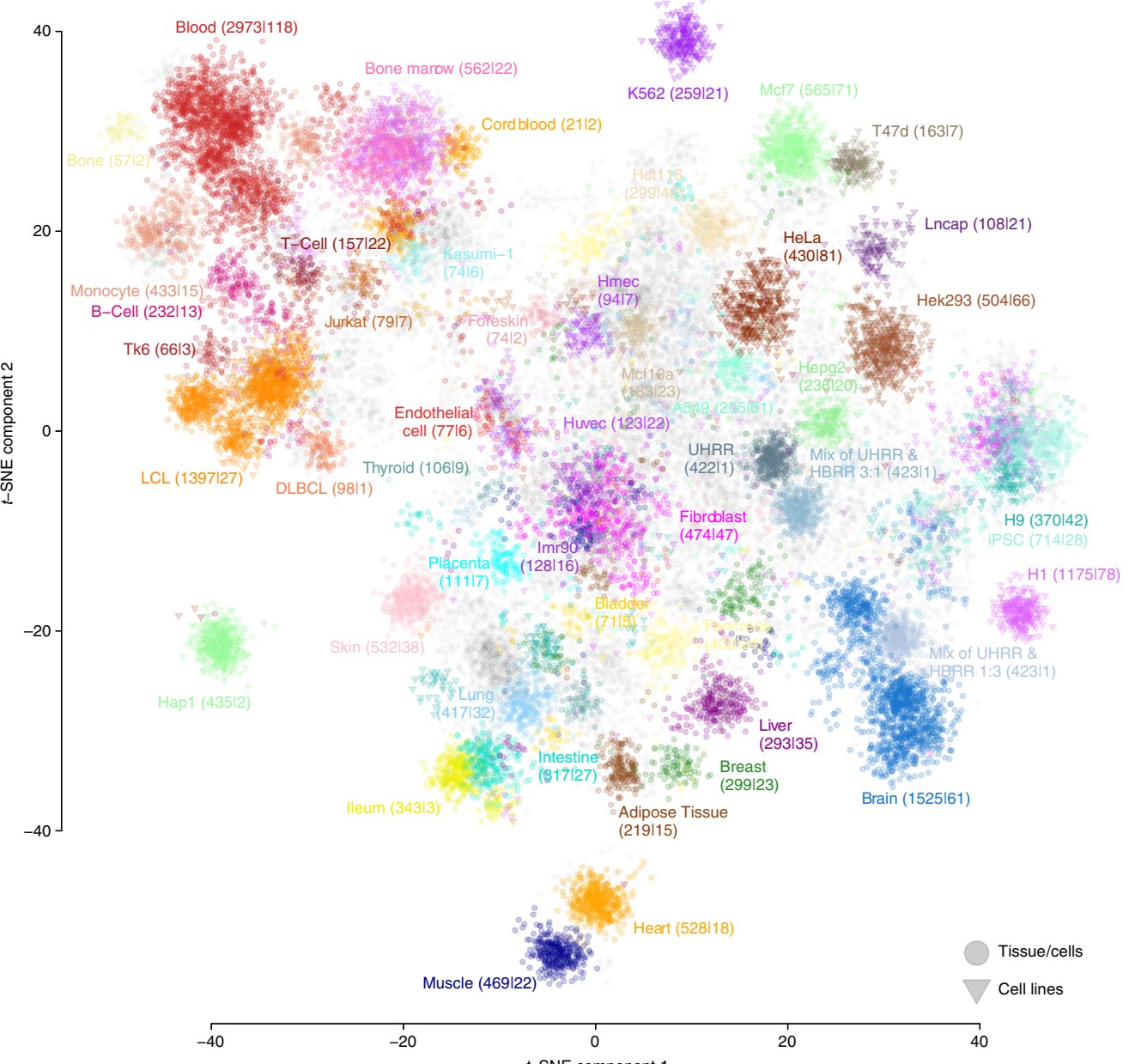

**Fig. 2** A compendium of gene expression profiles that can be used for gene function prediction. We downloaded 31,499 RNA-seq samples from ENA. These samples come from many different studies. They show coherent clustering after correcting for technical biases. Generally, samples originating from the same tissue, cell-type or cell-line cluster together. The two axes denote the two t-SNE components. The number of samples per tissue or cell-type are mentioned, and after the colon the number of unique studies is mentioned, indicating that samples cluster by tissue or cell-type, and that this clustering is not due to systematic technical confounding due to the fact that for a given tissue, samples come from only a single laboratory

**Table 1 Gene function prediction accuracy**

| Database | Number of gene sets | Gene sets ≥ 10 genes | Gene sets with significant predictive power | Median AUC |
|---|---|---|---|---|
| Reactome | 2,143 | 1,388 | 1,150 | 0.87 |
| GO molecular function | 4,070 | 726 | 398 | 0.82 |
| GO biological process | 11,753 | 2,576 | 1,115 | 0.82 |
| GO cellular component | 1,609 | 500 | 370 | 0.84 |
| KEGG | 186 | 186 | 168 | 0.84 |
| HPO | 7,920 | 3,281 | 1,887 | 0.73 |

*Note*: Gene co-expression information of 31,499 samples is used to predict gene functions. We show the prediction accuracy for gene sets from different databases
*AUC* area under the curve, *GO* gene ontology, *HPO* human phenotype ontology

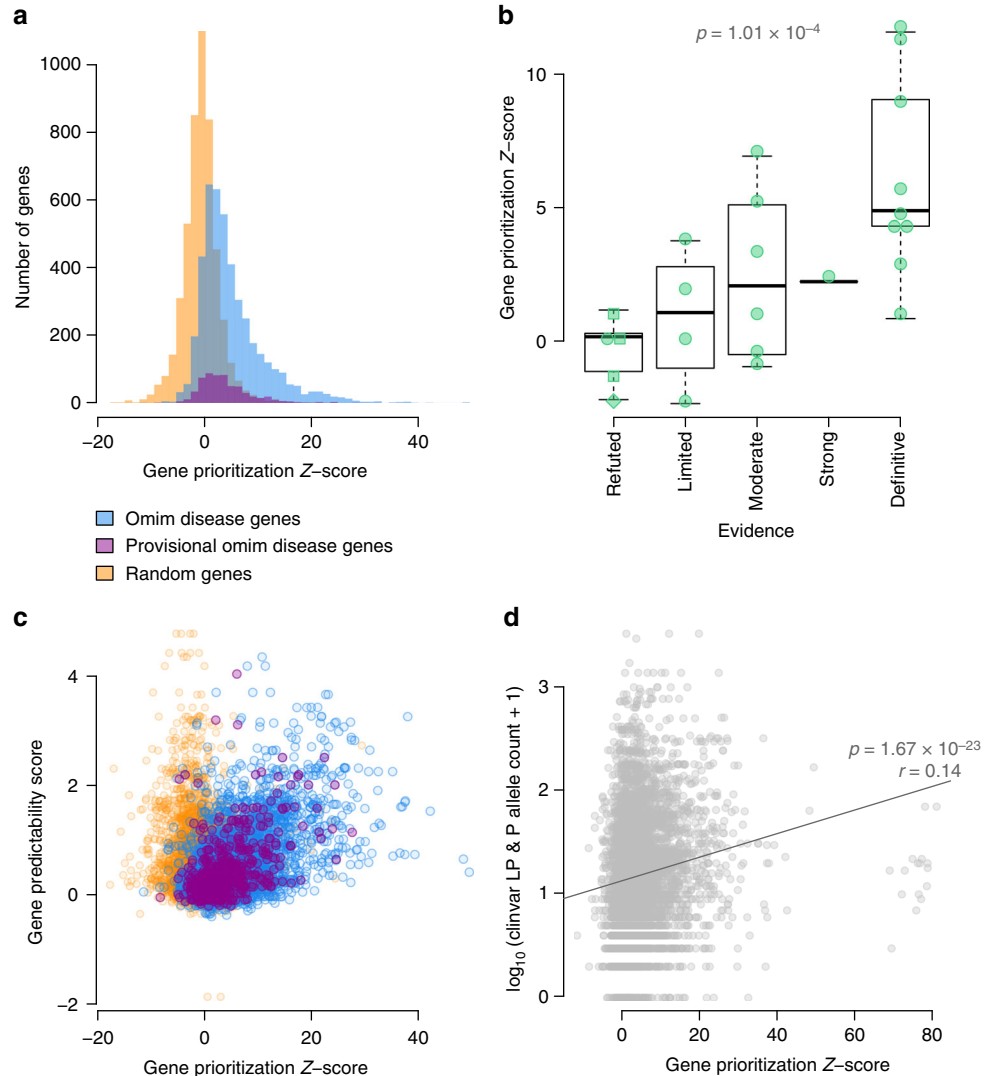

**Fig. 3** Performance of disease gene prioritization compared to random permutation. **a** OMIM disease genes and provisional disease genes have significantly stronger prioritization Z-scores compared to permuted disease genes (T-test p-values: $2.16 \times 10^{-532}$ and $5.38 \times 10^{-80}$, respectively). We also observe that the predictions of the provisional OMIM genes are, on average, weaker than the other OMIM disease genes (T-test p-value: $1.89 \times 10^{-7}$). Because we use a leave-one-out strategy when calculating prioritization Z-scores for genes that have already been associated to an HPO term, there is no prediction bias towards known associations. Therefore, this benchmark is informative of the power to predict novel associations (see methods). **b** We observe a significant relation (Spearman p-value: $1.01 \times 10^{-4}$) between the burden of evidence that a gene is associated to a disease and the GADO prioritization Z-score. Most genes are scored by[13] some additional refuted genes, denoted as squares or diamonds, are reported by ref. [8] and ref. [12] **c** We observe a clear relation between the prioritization Z-scores and the gene predictability scores (Pearson r = 0.54). We do not observe this relation in the permuted results. **d** Our gene prioritization Z-scores are significantly correlated (Pearson p-value: $1.67 \times 10^{-23}$) to the number of likely pathogenic (LP) and pathogenic (P) variants reported for a gene in ClinVar

**Gene predictability scores explain performance differences.** For some diseases in OMIM, GADO could not predict gene–phenotype combinations, as indicated by a prioritization Z-scores close to 0 or below 0 (Fig. 3a). For example, variants in SLC6A3 are known to cause infantile Parkinsonism-dystonia (MIM 613135)[24–26], but GADO was unable to predict the annotated HPO terms related to the Parkinsonism-dystonia for this gene. This may, however, be due to very low expression levels of SLC6A3 in most tissues except specific brain regions[27].

To better understand why we cannot predict HPO terms for all genes, we used the Reactome, GO and KEGG prioritization Z-scores. Jointly these databases comprise thousands of gene sets. Since these databases describe such a wide range of biology, we assumed that if a gene does not show any prediction signal for any gene set in these databases, gene co-expression is probably

not informative for this gene. To quantify this, we calculated, per gene, the average skewness of the pathway prioritization Z-score distribution of the Reactome, GO and KEGG gene sets. This average we use as the 'gene predictability score' for every gene that is independent of whether this gene is already known to play a role in any a disease or pathway (Fig. 3c, Supplementary Fig. 2). We then ascertained whether these 'gene predictability scores' are correlated with the HPO-based prioritization Z-score of the OMIM diseases, and found a strong correlation (r: 0.54, p-value: $1.14 \times 10^{-332}$) between the gene predictability scores and GADO's ability to identify a known disease gene (Fig. 3c, Supplementary Data 2).

**Prioritization of disease genes with limited evidence.** We used a set of disease genes that had been systematically studied by

Strande et al.[13] to ascertain the burden of evidence that exists for these genes, and complemented this list with a set of refuted genes[8, 12]. We observed that the GADO prioritization scores are related to this burden of evidence: refuted genes and genes with limited evidence have significantly lower prioritization Z-scores, compared to genes with more supporting evidence (Spearman p-value: $1.01 \times 10^{-4}$) (Fig. 3b). Our prioritization Z-scores are also correlated to the number of times an allele within a gene has been reported to be pathogenic or likely pathogenic in ClinVar[28] (r: 0.14 p-value: $1.67 \times 10^{-23}$) (Fig. 3d), which indicates that if many independent submissions have implicated the same gene in disease, that gene is more likely to be a true disease-causing gene. This is corroborated by the significant correlation between the ExAC missense constraint score[10] (a metric denoting a depletion of missense variation in a gene) and the number of submissions to ClinVar (r: 0.12 p-value: $8.81 \times 10^{-17}$) (Supplementary Fig. 4a). Interestingly, we do not observe a correlation between our prioritization Z-scores and the ExAC missense constraints (Supplementary Fig. 4b). A linear model to explain the number of ClinVar submissions using both our prioritization Z-scores together with the ExAC constraints performs significantly better than when solely using the ExAC constraints to predict the number of pathogenic or likely pathogenic in ClinVar (r: 0.21 vs. r: 0.12, ANOVA p-value: $1.24 \times 10^{-34}$). This indicates that GADO is informative for predicting the involvement of genes in disease, independent from ClinVar and ExAC.

A set of genes known to cause cardiomyopathy was scored for the amount of evidence in literature that these genes are involved in cardiomyopathy. Here, we again observe that genes with limited evidence have lower prioritization Z-scores (spearman p-value: $8.71 \times 10^{-04}$) (Supplementary Fig. 5), suggesting these could potentially reflect false-positive associations.

We were somewhat worried that such false-positive associations could detrimentally affect our gene–phenotype predictions. To ascertain this, we randomly added 10% more genes to each HPO-term and recalculated the predictions. We then observed that our predictions were robust, and that AUC values (indicating to what extent gene co-regulation can predict gene—phenotype associations) were very similar to the original AUC values (Pearson correlation r = 0.97, Supplementary Fig. 5).

**Benchmarking GADO using cases with realistic phenotyping**. Although these in silico benchmarking demonstrated the potential of GADO, it used all annotated HPO terms for a disease. In practice, however, patients may only present with a limited number of the annotated features of a disease. To perform a validation that was a realistic reflection of clinical practice, we used exome sequencing data of 83 patients with a known genetic diagnosis. We used their phenotypic features as listed in their medical records prior to when the genetic diagnosis had been made (Supplementary Table 1). Per patient, our exome-sequencing pipeline GAVIN[29] returned a median of 55 possible disease-causing genes with variants that are rare and predicted to be deleterious (Supplementary Data 3). We then ran GADO and observed that for 41% of these patients the actual causative gene ranked in the top 3 (median rank was 6.5 for all 83 patients, Supplementary Fig. 6). Using a stringent threshold (prioritization Z-score ≥ 5), which we also used for the prioritization of unsolved cases (see below), to select strong candidate genes, we identified the causative gene for 17 cases (20%) while only needing to follow-up a single variant (range 0–5) per patient on average.

Because of our leave-one-out procedure when calculating prioritization Z-scores for known disease genes (see methods), our performance in solved cases is indicative of the power of GADO to prioritize novel disease-associated genes without prior annotations or associations. However, these unbiased predictions can sometimes cause problems when using GADO in clinical practice, because GADO cannot predict every known gene-HPO combination accurately. As such some of these known gene-HPO combinations might have rather insignificant Z-scores. To make sure GADO is also suited for cases with variants in currently known disease associated genes, we adjusted our prediction matrix to ensure that known HPO-term associations for genes are also prioritized (see methods). This does not affect GADO's ability to prioritize novel diseases genes, but solely helps the prioritization performance of known disease genes, but ensures that users of the GADO website will see these known disease-phenotype as top-ranking genes. By doing this we achieved a similar prioritization performance as compared with Exomiser (Supplementary Table 2, Fig. 4a). For this comparison, we used both methods to rank the on average 663 variants that are selected by Exomiser. For Exomiser, we used the default 'combined prioritization' strategy that is based on the variant score and the gene score, whereas in GADO we solely used the prioritization Z-scores (Supplementary Methods 2). Although our median rank of the causative gene is better compared with Exomiser (GADO: 12.5 vs. Exomiser: 21), Exomiser on the other hand, is able to rank more genes in the top 3 (Exomiser: 28 vs. GADO: 14).

**Clustering of HPO terms**. In addition to ranking potentially causative genes based on a patient's phenotype, GADO can be used to cluster HPO terms based on the genes that are predicted to be associated to these HPO terms. This can help to identify pairs of symptoms that often occur together, as well as symptoms that rarely co-occur. In a patient diagnosed with a glycogen storage disease, GSD type Ib, caused by compound heterozygous variants in *SLC37A4* (MIM 602671) and Dilated Cardiomyopathy (DCM) that is probably caused by a truncating variant in *TTN* (MIM 188840) HPO terms related to GSD type Ib ('leukopenia' (HP:0001882) and 'inflammation of the large intestine' (HP:0002037)) cluster together, while Cardiomyopathy (HP:0001638) was only weakly correlated to these specific features (Fig. 4b).

**Reanalysis of previously unsolved cases**. To assess GADO's ability to discover previously unknown disease genes, we applied it to data from 61 patients who are suspected to have a Mendelian disease but who did not receive a genetic diagnosis. All patients had undergone prior genetic testing (WES with analysis of a gene panel according to their phenotype, Supplementary Table 3). On average GADO reported 2.9 genes with a prioritization Z-score ≥ 5 (which we used as an arbitrary cut-off and that corresponds to a p-value ≤ $5.7 \times 10^{-7}$) and which were further assessed. In ten cases, we identified variants in genes not associated to a disease in OMIM or other databases, but for which we could find literature or for which we gained functional evidence implicating their disease relevance (Table 2). For example, we identified two cases with DCM with rare compound heterozygous variants in the *OBSCN* gene (MIM 608616) that are predicted to be damaging. In literature, inherited variant(s) in *OBSCN*, encoding obscurin, are associated with hypertrophic CM[30] and DCM[31]. Furthermore, obscurin is a known interaction partner of titin (TTN), a well-known DCM-related protein[30]. Another example came from a patient with ichthyotic peeling skin syndrome, which is caused by a damaging variant in *FLG2* (MIM 616284). We recently published this case where we prioritized this gene using an alpha version of GADO[32].

We compared GADO with Exomiser, ENDEAVOR[33], and ToppGene[34] on our unsolved cases for which we identify a

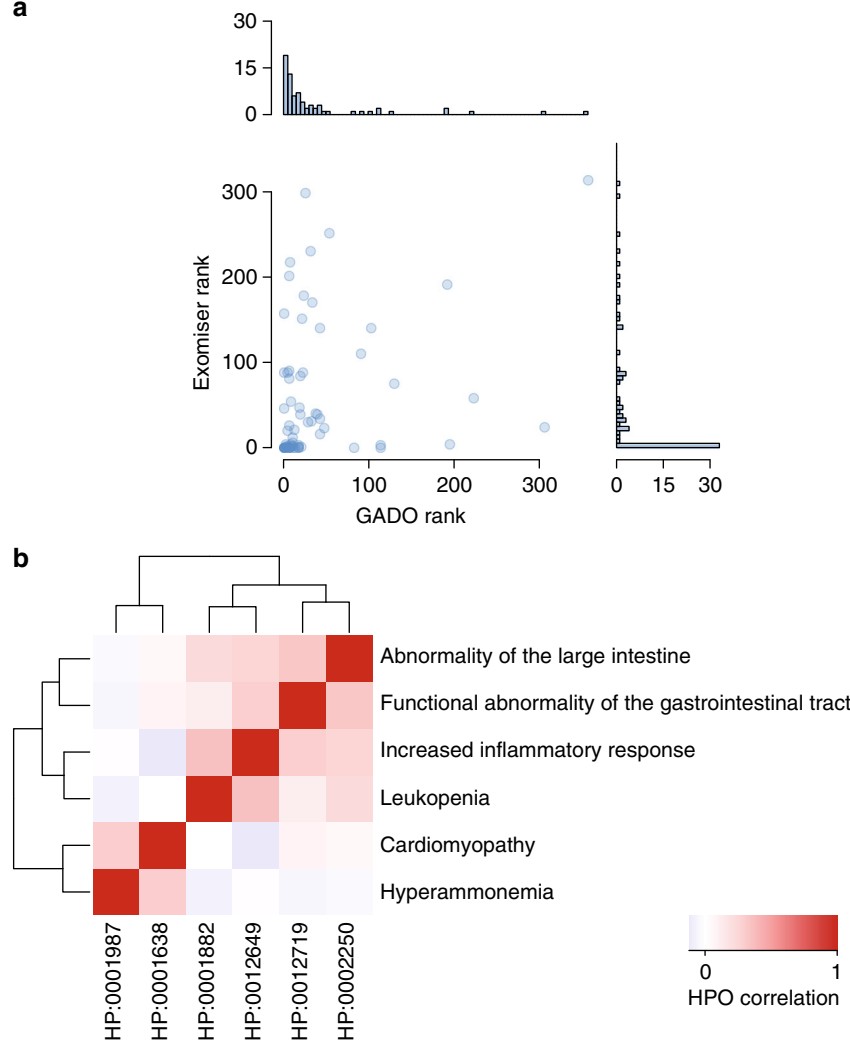

**Fig. 4** Performance of GeneNetwork on solved cases. **a** Comparison between using GADO and Exomiser to rank candidate variants. **b** Our cohort contained a case with two distinct conditions, and clustering showed the HPO terms of the same disease are closest to each other. Note, the HPO term "Inflammation of the large intestine" did not yield a significant prediction profile and therefore the parent terms "Abnormality of the large intestine", "Increased inflammatory response" and "Functional abnormality of the gastrointestinal tract" were used for this case

strong candidate (Supplementary Note 2). Exomiser could be run directly using the HPO terms. The other tools required a list of training genes (i.e., the genes known to cause a specific HPO term), but provided no options to integrate the results of multiple sets of training genes. We therefore only used ENDEAVOR and ToppGene for those cases with a single reported HPO term. ENDEAVOR supported a maximum of 200 input genes in the training set (i.e., those genes known to cause a specific HPO term) and at most 200 genes to prioritize (i.e., those genes in which rare variants had been observed). If for an HPO term over 200 genes were known, we selected a random subset of 200 genes. If a patient had candidate variants in more than 200 genes, we trimmed this set to 200 genes by randomly removing genes, while ensuring that the known causative gene was retained. The median rank of these genes was 3 for GADO, 68.5 for Exomiser, 7.5 for ENDEAVOR and 24 for ToppGene (Supplementary Data 4). The Exomiser ranks however, are not directly comparable since Exomiser does its own variant select which yields more variants than GAVIN, the method we used prior to running GADO, ENDEAVOR and ToppGene. To overcome this, we also calculated the percentile of the candidate gene among the total genes selected either by GAVIN or Exomiser, the median percentile for GADO was: 1.2 and for Exomiser: 7.9.

**GADO webserver & standalone command line**. All analyses described in this paper can be performed using our online toolbox at www.genenetwork.nl. Users can perform gene prioritizations using GADO by providing a set of HPO terms and a list of candidate genes (Fig. 5a). We have also made a standalone command line version of GADO that can easily be integrating in a bioinformatics pipeline (https://github.com/molgenis/systemsgenetics/wiki/GADO-Command-line). Per gene, it is also possible to download all prioritization Z-scores for the HPO terms and pathways. Furthermore, the predicted pathway and HPO annotations of genes can be used to perform function enrichment analysis (Fig. 5b). We also support automated queries to our database using a http+JSON api.

**Discussion**

The identification of new disease-causing genes is a daunting process. GADO can aid in the discovery of these unknown disease genes. The main advantage of our methodology is that it does not rely on any prior knowledge about the genes that we prioritize

**Table 2 Unsolved cases with new candidate genes**

| HPO terms used | Number of genes with candidate variant | Number of genes with Z ≥ 5 | Candidate gene | Variants | CADD scores | GnomAD minor allele frequency | Supporting papers | Expression in relevant tissue |
|---|---|---|---|---|---|---|---|---|
| HP:0001644 | 215 | 5 | *OBSCN* | NM_001098623.2:c. [15037 C > T]; [20963delC] | 24.8 25.2 | $8.0 \times 10^{-5}$ $1.7 \times 10^{-3}$ | 30, 31 | Yes |
| HP:0001644 | 226 | 3 | *OBSCN* | NM_001098623.2:c. [5545 C > T]; [22384 + 3 _22384 + 21del] | 14.7 7.8 | $3.2 \times 10^{-4}$ 0 | 30, 31 | Yes |
| HP:0008066 HP:0008064 | 359 | 3 | *FLG2* | NM_001014342.2:c. [632 C > G];[632 C > G] | 35.0 | $1.1 \times 10^{-5}$ | 49 | Yes |
| HP:0001263 HP:0001249 HP:0000717 HP:0000708 HP:0002167 HP:0002360 HP:0000664 | 206 | 12 | *INO80* | NM_017553.2:c. [898C > T] | 34 | 0 | 50, 51 | Yes |
| HP:0001644 | 120[a] | 2 | MB | NM_00203377.1:c. [214 G > A] | 22.4 | $3.6 \times 10^{-5}$ | 52 | Yes |
| HP:0001644 | 120[a] | 1 | *SYNPO2L*[b] | NM_001114133.2:c. [473 G > A] | 24.1 | $5.4 \times 10^{-4}$ | 53 | Yes |
| HP:0001638 | 292 | 4 | *NRAP*[b] | NM_001261463.1:c. [4648 C > T] | 20.4 | $8.7 \times 10^{-4}$ | 54 | Yes |
| HP:0004322 HP:0001249 | 381 | 10 | *CCNB2* | NM_004701.3:c. 25-3_25delCAGG | 24.5 | 0 | 55 | Yes |
| HP:0003493 HP:0002583 | 246 | 6 | *LY75* | NM_002349.2: c. 3476 C > T(;) 23 C > G | 22.7 24.1 | $3.2 \times 10^{-3}$ $2.6 \times 10^{-3}$ | 56 | Yes |
| HP:0012649 HP:0002583 HP:0001890 | 318 | 8 | *AGAP2* | NM_001122772.1:c. 421delC | 27.2 | 0 | 57 | Yes |

*Note*: In 10 out of 61 unsolved patients we identified likely causative genes that were previously unknown. For these genes we found literature that indicates these genes fit the phenotype of these patients or we gained functional evidence implicating their disease relevance. HP:0001644 = Dilated cardiomyopathy; HP:0008066 = Abnormal blistering of the skin; HP:0008064 = Ichthyosis; HP:0001263 = Global developmental delay; HP:0001249 = Intellectual disability; HP:0000717 = Autism; HP:0000708 = Behavioral abnormality; HP:0002167 = Neurological speech impairment; HP:0002360 = Sleep disturbance; HP:0000664 = Synophrys; HP:0001638 = Cardiomyopathy; HP:0004322 = Short stature; HP:0001249 = Intellectual disability; HP:0003493 = Antinuclear antibody positivity; HP:0002583 = Colitis; HP:0012649 = Increased inflammatory response; HP:0001890 = Autoimmune hemolytic anemia
[a]These variants were pre-filtered for family segregation
[b]The variants in these genes do not fully explain the phenotype but are likely contributing to the phenotype

and can therefore also detect genes for which nothing is known. Instead, we used predicted gene functions based on co-regulation networks extracted from a large compendium of publicly available RNA-seq samples allowing accurate expression quantification of many genes, including lowly expressed genes and non-coding genes[18]. A realistic benchmark using real cases and features listed in the medical records allowed us to identify the causative genes for 20% of the cases, while only requiring us to follow-up on average only a single gene per patient.

GADO is trained in such a way that for each gene–phenotype combination that is already known, this knowledge is not used when using co-regulation information to make inferences on that specific gene–phenotype association. A major advantage of this is that our gene–phenotype predictions are not biased towards known associations. However, since we do not incorporate these known disease associations into our model, the performance of GADO is lower when studying patients with mutations in well-established genes, as compared to methods that explicitly use these known gene–phenotype associations. To accommodate this issue, we have added these known gene–phenotypes to GADO, to ensure GADO users will not miss out on known associations.

This is useful for genes with a low predictability score indicating that gene expression data is not informative for its function predictions and for genes such as *TTR* that act in a unique manner compared to other genes that give rise to CM. *TTR* is implicated in hereditary amyloidosis (MIM 105210)[35] and there is a large amount of evidence linking this gene to CM. Mutations in *TTR* cause accumulation of the transthyretin protein in different organ systems, including the heart, resulting in CM. However, this gene is primarily expressed in the liver. Therefore, its disease mechanism is different from other mechanisms resulting in CM, as many inherited CMs are caused by deleterious variants in genes highly expressed in the heart and directly affecting the function of the cardiac sarcomere[36]. Because this gene is expressed in a different tissue than all other CM genes, co-expression is not informative and as a result the phenotypic function prediction for this gene is worse than we would expect based on the predictability score.

Finally, we used GADO on 61 unsolved cases and identified for 10 cases (16.4%) potential disease genes that are strong candidates based on literature or functional evidence. All these samples already went through an extensive diagnostic procedure so these

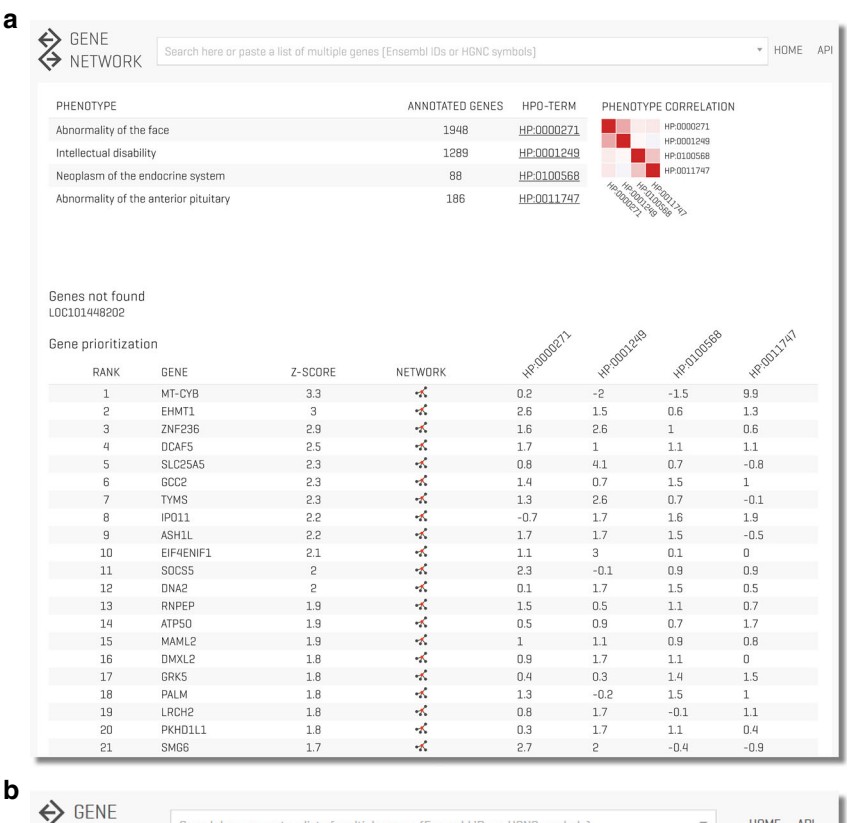

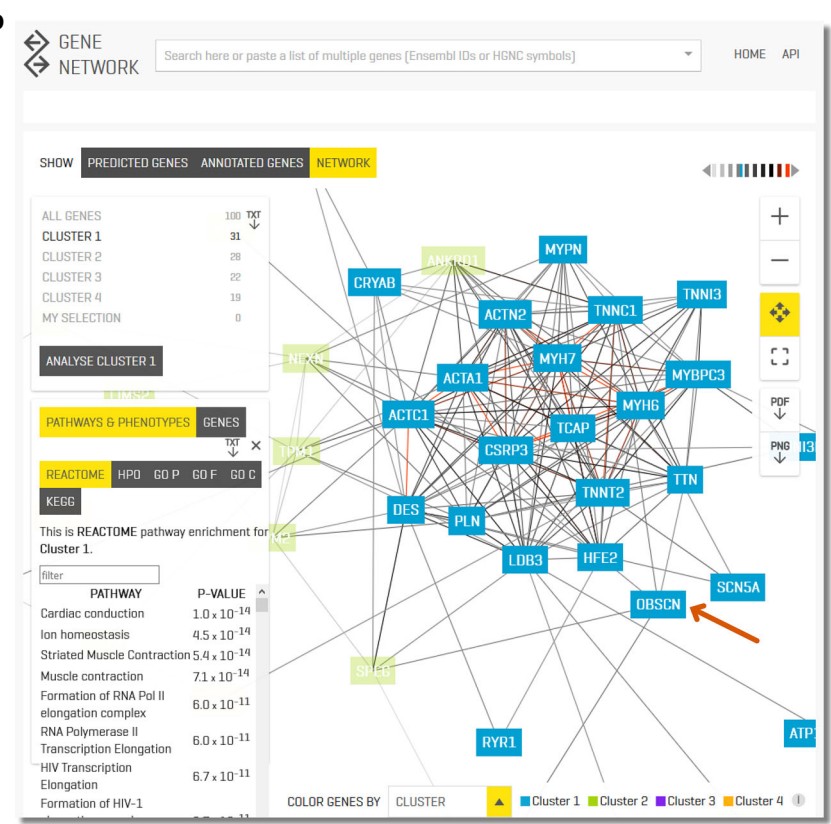

**Fig. 5 a** Prioritization results of one of our previously solved cases (www.genenetwork.nl). This patient was diagnosed with Kleefstra syndrome. The patient only showed a few of the phenotypic features associated with Kleefstra syndrome and additionally had a neoplasm of the pituitary (which is not associated with Kleefstra syndrome). Despite this limited overlap in phenotypic features, GADO was able to rank the causative gene (EHMT1) second. Here, we also show the value of the HPO clustering heatmap: the two terms related to the neoplasm cluster separately from the intellectual disability and the facial abnormalities that are associated to Kleefstra syndrome. **b** Clustering of a set of genes allowing function/HPO enrichment of all genes or specific enrichment of automatically defined sub clusters. Here, we loaded all known DCM genes and OBSCN, and we focus on a sub-cluster of genes containing OBSCN (highlighted by the arrow). We see that it is strongly co-regulated with many of the known DCM genes. Pathway enrichment of this sub-cluster reveals that these genes are most strongly enriched for the muscle contraction Reactome pathway. DCM, Dilated Cardiomyopathy

findings are on top of the normal diagnostic yield. When applying GADO, we could identify a very likely causative gene for 16.4% of these unsolved cases, based on the existence of circumstantial evidence in literature on these genes. This is only a bit lower than what we observed for solved cases where the causative gene is known: when we assumed that the causative gene was not yet known, GADO identified the causative gene for 20% of the cases. We should note that this 16.4% yield in unsolved cases might actually be an underestimate: GADO also had prioritized genes with a high prioritization Z-score for some of the other unsolved cases, of which some are likely to be responsible for the phenotypes observed in these patients. Regretfully, for these genes no literature currently exists that supports their role in the symptoms of these patients. This is one of the pertinent issues when it comes down to diagnosing patients. Additional repositories that use orthogonal data to make inferences on the phenotypic consequences of mutations in genes and initiatives like Genematcher[37] therefore remain urgently needed, in order to increase the diagnostic yield.

Given that nearly 5% of patients with a Mendelian disease have another genetic disease[38], it is important to consider that multiple genes might each contribute to specific phenotypic effects. Clinically, it can be difficult to assess if a patient suffers from two inherited conditions, which may hinder variant interpretation based on HPO terms. We showed that GADO can disentangle the phenotypic features of two different diseases manifesting in one patient by correlating and subsequently clustering the profiles of HPO terms describing the patient's phenotype. If the HPO terms observed for a patient do not correlate, it is more likely that they are caused by two different diseases. An early indication that this might be the case for a specific patient can simplify subsequent analysis because the geneticist or laboratory specialist performing the variant interpretation can take this in consideration. GADO also facilitates separate prioritizations on subsets of the phenotypic features.

We compared GADO to Exomiser, which is closely related to GADO as it prioritizes genes based on specified HPO terms and also infers HPO annotation for unknown genes[6]. The gene prioritization by Exomiser is based on the effects of orthologous in model organisms and applies a guilt-by-association method using protein-protein associations provided by STRING[39]. Exomiser performs similar to GADO in ranking known disease-causing genes (Supplementary Fig. 7, Supplementary Table 2) and is also able to identify potential new genes in human disease. However, only a subset of the protein-coding genes have orthologous genes in other species for which a knockout model also exists and the used STRING interactions are biased towards well-studied genes and rely heavily on existing annotations to biological pathways (Supplementary Fig. 8). There are however, still 3922 protein-coding genes that are not currently annotated in any of the databases we used, and there are even more non-coding genes for which the biological function or role in disease is unknown. Since GADO does not rely on prior knowledge, it can be used to prioritize variants in both coding and non-coding genes (for which no or limited information is available). GADO thus enables the discovery of novel human disease genes and can complement existing tools in analyzing the genomic data of patients who have a broad spectrum of phenotypic abnormalities.

Other tools such as ENDEAVOR[33], ToppGene[34], and Suspects[40], that have been used successfully before to prioritize candidate genes are not directly comparable to GADO, since these tools work by either supplying a single HPO term or a set of training genes. However, these tools can be used to successfully prioritize disease genes[41]. In some cases, a single HPO term might be sufficient or a custom gene can be useful when a specific syndrome is suspected and several other genes have already been implicated for this syndrome. Unfortunately, in clinical practice often multiple HPO terms are needed to describe a patient's phenotype (e.g., for our set of solved cases we used two HPO terms on average). Moreover, it is also often unclear which syndrome a patient has, which inhibits the ability to prioritize genes based on already associated genes to a syndrome.

We found that for some disease genes GADO is unable to predict the already known phenotypic consequences. This is partially explained by genes for which gene-expression data is not informative for function predictions. For instance, because a gene has very low gene expression, because different splice variants have different functions, or because the regulation of a gene its function relies heavily on post-translation modification. We have defined an empirical measurement called 'gene predictability' that indicates how informative gene expression is for function prediction of individual genes. We found a strong correlation between this predictability metric and our ability to predict known phenotypic consequences of disease associated genes. This however does not fully explain our inability to predict known phenotypic consequences, in some cases this can simply be due to an alternative disease mechanism.

GADO can also point to genes that may have been falsely associated to a disease. Genes for which there is limited evidence to link them to a disease have, on average, lower prioritization Z-scores compared to well established genes and genes that have been refuted in literature have even lower scores. In addition, we found a statistically significant association between the prioritization Z-scores of known disease-gene combinations and the number of pathogenic or likely pathogenic alleles reported in ClinVar, thereby assuming that the genes with many submissions are more likely to be truly related to human disease. We also observed a statistically significant correlation between the ExAC missense constraint and the number of alleles submitted to ClinVar. Interestingly, the ExAC missense constraints are not correlated to our prediction scores showing that both can be used as independent predictors of potential false-positive disease associations.

The median prediction performance of HPO terms is lower compared to the other gene sets databases used in our study, such as Reactome. This may be due to the fact that phenotypes can arise by disrupting multiple distinct biological pathways. For instance, DCM can be caused by variants in sarcomeric protein genes, but also by variants in calcium/sodium handling genes or by transcription factor genes[36]. As our methodology makes guilt-by-association predictions based on whether genes are showing correlated gene expression levels, the fact that multiple separately working processes can cause the same phenotype can reduce the accuracy of the predictions (although it is often still possible to use these predictions, e.g., the DCM HPO phenotype prediction performance $AUC = 0.76$). We envision that by creating sub-clusters, based on these different pathways, and redoing our gene-expression based predictions, it might be possible to further improve the performance of HPO based prioritizations in the future. Insufficient statistical power might also hinder accurate predictions for HPO terms. This may specifically be true for genes that are poorly expressed or expressed in only a few of the available RNA-seq samples. The latter issue we expect to overcome in the near future as the availability of RNA-seq data in public repositories is rapidly increasing. Initiatives such as Recount[42] or SkyMap[43] enable easy analysis on these samples, allowing us to update our predictions in the future, thereby increasing our prediction accuracy.

We have developed GADO, a method that can aid users in prioritizing genes using multiple patient-specific HPO terms. We performed our GADO benchmarking while using GAVIN for the selection of genes that contain (likely pathogenic) rare variants.

However, GADO can work with any other methodology for identification of genes harboring rare and potentially pathogenic alleles. GADO prioritizes variants in coding and non-coding genes, including genes for which there is no current knowledge about their function and those that have not been annotated in any ontology database. This gene prioritization is based on co-regulation of genes identified by analyzing 31,499 publicly available RNA-seq samples. Therefore, in contrast to many other existing prioritization tools, GADO has the ability to identify novel genes involved in human disease. By providing a statistical measure of the significance of the ranked candidate variants, GADO can provide an indication for which genes its predictions are reliable. GADO can also detect phenotypes that do not cluster together, which can alert users to the possible presence of a second genetic disorder and facilitate the diagnostic process in patients with multiple non-specific phenotypic features. GADO can easily be combined with any filtering tool to prioritize variants within WES or WGS data and can also be used in gene panels such as PanelApp[44]. Finally, GADO can aid in the identification of genes falsely associated to diseases. GADO is freely available at www.genenetwork.nl and https://github.com/molgenis/systemsgenetics/wiki/GADO-Command-line to help guide the differential diagnostic process in medical genetics.

## Methods

**Gene co-regulation and function predictions**. We used publicly available RNA-seq samples from the European Nucleotide Archive (ENA) database[45] to predict gene functions and gene-HPO term associations. After processing and quality control we included 31,499 samples for which we have expression quantification on 56,435 genes (in-depth details are provided in Supplementary Methods 1). We subsequently performed a PCA on the gene correlation matrix and selected 1,588 reliable principal components (PCs) (Cronbach's Alpha ≥ 0.7).

We used the eigenvectors of these 1,588 PCs to predict gene functions using a method we published earlier[46] (Fig. 1). Per PC we used the eigenvector coefficients for the genes that are part of a gene set and the eigenvector coefficients of the background genes that are not in the current gene set. We used a student's $T$-test to compare the eigenvector coefficients of the genes in the gene set to the eigenvector coefficients of the background genes. We then calculated a $T$-test $p$-value and converted this to a $Z$-score. This resulted in a matrix where for each gene set for each of the 1,588 PCs a $Z$-score had been calculated. These $Z$-scores reflect the importance of a specific component for predicting which genes are part of a specific gene set. In order to finally predict which genes are part of a specific gene set, we calculate the correlation between the 1,588 $T$-test $Z$-scores for a given gene set and the 1,588 eigenvector coefficients of each gene. The rationale here is that if the same components are relevant for an individual gene (as determined through the eigenvector coefficients) and also for a specific pathway (large $Z$-score from the $T$-test) then this indicates that the expression regulation of that gene is similar to the expression regulation pattern of that pathway. The $p$-value that belongs to this correlation was subsequently transformed to a $Z$-score and was used as the prioritization $Z$-score (where a high score makes it more likely that a gene is part of a gene set).

**Leave-one-out procedure**. However, there is one exception to this procedure when we want to calculate the prioritization $Z$-score for a gene—gene set combination when that gene—gene set is already known: If we would include this gene when conducting the 1,588 $T$-tests and subsequent $Z$-scores (for determining the importance of each component when predicting this gene set), a positive correlation between the 1,588 eigenvector coefficients and the 1,588 $Z$-scores is expected, which leads to a bias in the predictions towards genes with a known HPO annotation. To prevent this bias, we used a leave-one-out procedure where we always exclude the current gene from the gene set and recalculate the $Z$-scores derived from the $T$-tests before correlating the profile of a gene set to the eigenvector coefficients of this gene. This ensures that there is no inflation of prioritization $Z$-score for genes that already have been annotated to the corresponding gene set. It also allows use to calculate reliable AUC based on the current annotations to a gene set[46].

To determine the accuracy of our predictions, we assessed our ability to predict known gene set annotations: for each gene set, we calculated an Area Under the Curve (AUC) using the prioritization $Z$-scores of the genes that are part of a set versus those that are not part of a set. We used a Mann–Whitney $U$ test to calculate if the prioritization $Z$-score of currently annotated genes are significantly larger than the genes not annotated to this gene set. If this is not the case, we concluded that we could not make meaningful prioritizations for this gene set by using the 1588 principal components.

We applied this methodology to the gene sets described by terms in the following databases: Reactome and KEGG pathways, Gene Ontology (GO) molecular function, GO biological process and GO cellular component terms and finally to HPO terms. We excluded gene sets with fewer than 10 annotated genes and with a $p$-value ≤ 0.05 (Bonferroni corrected for the number of pathways in a database).

**Gene predictability scores**. To explain why for some genes we cannot predict known HPO annotation, we have established a gene predictability score. We have calculated this gene predictability using the prioritization $Z$-scores based on Reactome, GO and KEGG. For each gene and for each database we calculated the skewness in the distribution of the pathway prioritization $Z$-scores of the gene sets. We used the average skewness as the gene predictability score.

**GADO predictions**. To identify potential causative variants in patients, we used HPO terms to describe a patient's features. We only used the HPO terms which have significant predictive power (Bonferroni corrected $p$-value of $U$ test to calculate the AUC ≤ 0.05). If the predictions for a patient's HPO term were not significant, the parent/umbrella HPO terms were used (Supplementary Fig. 1). The online GADO tool suggests the parent terms from which the user can then select which terms should be used in the analysis. The gene prioritization $Z$-scores for an HPO term were used to rank the genes. If a patient's phenotype was described by more than one HPO term, a meta-analysis was conducted to integrate the predictions of the used HPO terms. In these cases a combined prioritization $Z$-score was calculated using the $Z$-transform test[47]. This was done by adding the prioritization $Z$-scores for each of the patient's HPO terms and then dividing by the square root of the number of HPO terms. This will result in a combined prioritization $Z$-score reflecting the predictions of all the supplied HPO terms. The genes with the highest prioritization $Z$-scores are predicted to be the most likely candidate causative genes for a case.

In addition to the predictions described above, we have created a GADO option which ensures any HPO term associated to a gene obtains a minimum prioritization $Z$-score of 3 for this gene. This option is not used for the benchmark results shown within this manuscript with the exception of the comparison against Exomiser using previously solved cases which was ran once with, and once without this option.

Gene prioritization analysis using HPO terms and a list of candidate genes can be performed at https://www.genenetwork.nl.

**Validation of disease-gene predictions**. To benchmark our method we used the OMIM morbid map[23] downloaded on March 26, 2018, containing all disease-gene-phenotype entries. From this list, we extracted the disease-gene associations, excluding non-disease and susceptibility entries. We extracted the provisional disease-gene associations separately. For each disease in OMIM, we used GADO to determine the rank of the causative gene among all genes in the OMIM morbid map. For this we used all phenotypes annotated to the OMIM disease. If any of the HPO terms did not have significant predictive power, the parent terms were used.

To determine if these distributions were significantly different from what we expect by chance, we permuted the data. We replaced the existing gene-OMIM annotation but assigned every gene to a new disease (keeping the phenotypic features for a disease together), assuring that the randomly selected gene was not already annotated to any of the phenotypes of the original gene.

**Cohort of previously solved cases**. Whole exome sequencing was performed in all patients in accordance with the regulations and ethical guidelines of the University Medical Center Groningen (UMCG Medical Ethics Committee). To test if GADO could help prioritize genes that contain the causative variant, we used 83 samples of patients who were previously genetically diagnosed through whole exome analysis or gene panel analysis. These samples encompass a wide variety of different Mendelian disorders (Supplementary Table 1). To assess which genes harbor potentially causative variants, we first annotated the variants from the exome sequencing using GAVIN. For 11 of the previously solved cases, GAVIN did not flag the causative variant as a candidate. Since this is the result of the specificity and sensitivity tradeoff made by GAVIN, we added the causative genes that had been missed by GAVIN for these 11 cases, so that we could still benchmark GADO on these patients.

The phenotypic features of a patient were translated into HPO terms, which were used as input to GADO. Here, we only used features reported in the medical records prior to the molecular diagnosis. If any of the HPO terms did not have significant predictive power, the parent terms were used. From the resulting list of ranked genes, the known disease genes harboring a potentially causative variant were selected. Next, we determined the rank of the gene with the known causative variant among the selected genes. If a patient harbored multiple causative variants in different genes, the median rank of these genes was reported (Supplementary Table 1).

**Unsolved cases cohorts**. In addition to the patients with a known genetic diagnosis, we tested 61 unsolved cases (Supplementary Table 2). These are patients with mainly cardiomyopathies or developmental delay. All patients were previously

investigated using exome sequencing, by analyzing a gene panel appropriate for their phenotype. To allow discovery of potential novel disease genes, we used GADO to rank genes with candidate variants that are identified using GAVIN. For genes with a prioritization Z-score ≥ 5, a literature search for supporting evidence was performed to assess whether these genes are likely candidate genes.

**Variant calling and processing of benchmark samples**. The solved and unsolved samples were processed in the following manner. For variant calling, we used the available WES or WGS data from patients with and without genetic diagnosis. These samples were genotyped using a relatively standard BWA and GATK pipeline. For a detailed description of the genotype pipeline see: https://molgenis.gitbooks.io/ngs_dna/ (version 3.4.0). For the WGS samples, we confined our analysis to the exome. For variant annotation, we used GAVIN to annotate our variants to obtain a list of candidate variants. GAVIN prioritizes genes based on, among other factors, minor allele frequency and gene-recalibrated CADD scores (for details see[29]).

**Comparing GADO and Exomiser on cases with known disease genes**. To evaluate GADO's performance, we compared GADO with Exomiser[48] (version 10.1.0, with exomiser-phenotype-1802 and exomiser-genome-hg19–1805 files from https://data.monarchinitiative.org/exomiser/data/). Both GADO and Exomiser were given each patient candidate gene list along with their respective set of phenotypes as input. Default settings were used. We used the gene rankings based on "EXOMISER_GENE_COMBINED_SCORE" and identified the rank of the causative gene (Supplementary Table 2). In case of a tie, the average rank of the ties was reported. If a patient harbored multiple causative variants, the median rank of the genes harboring the causative variants was reported. To ensure a fair comparison, we used GADO on the set of genes reported by Exomiser (Supplementary Table 2).

**Website**. To make our method and data available, we have developed a website available at www.genenetwork.nl that can be used to run GADO, lookup gene functions predictions, visualize networks using co-regulations scores and perform function enrichments of sets of genes (Supplementary Note 3).

**GADO prediction of false positives**. Gene confidence annotations were retrieved from previous studies[13]. We used annotations from[13] in our figure. We added an additional 4 genes from to the refuted category as the variants associated to the diseases have been found to be to common[8, 12]. We assigned a score of 1 to the refuted genes, 2 to limited genes, 3 to moderate genes, 4 to strong genes, and 5 to definitive genes. Next, we calculated the spearman-rank correlation between these values and the prioritization Z-scores for the corresponding genes (Fig. 3b).

## Data availability

The RNA-seq data used in this study is available at the European nucleotide archive (https://www.ebi.ac.uk/ena), the included samples are listed in Supplementary Data 1. The matrices needed for the command line version of GADO are hosted on figshare and are listed at https://github.com/molgenis/systemsgenetics/wiki/GADO-Command-line. Due to the nature of the consent given by individuals, we are not allowed to share the exome sequencing data of the solved and unsolved cases. In Supplementary Data 3 we have listed the genes harboring candidate variants that are identified by GAVIN to allow reproducibility of our results. The matrix with gene prioritization z-scores for HPO-terms is available here: https://doi.org/10.6084/m9.figshare.8144291.

## Code availability

The source code of GADO can be found here: https://github.com/molgenis/systemsgenetics/wiki/GADO-Command-line

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

## Acknowledgements

We are grateful for the participation of the patients and their parents in this study. We thank Kate Mc Intyre for editing the manuscript and Marieke Bijlsma, Gerben van der Vries, Sido Haakma and Pieter Neerincx for support with the computational analyses. This work was carried out on the computer cluster of the Genomics Coordination Center, hosted at the University of Groningen Center for Information Technology (Strikwerda, W. Albers, R. Teeninga, H. Gankema and H. Wind) and Target storage (E. Valentyn and R. Williams). Target is supported by Samenwerkingsverband Noord Nederland, the European Fund for Regional Development, the Dutch Ministry of Economic Affairs, Pieken in de Delta and the provinces of Groningen and Drenthe. This work is supported by a grant from the European Research Counsil (ERC Starting Grant agreement number 637640 ImmRisk) to Lude Franke and two VIDI grants (917.14.374 and 917.16.455) from the Netherlands Organisation for Scientific Research (NWO) to Lude Franke and Morris Swertz. This work was supported by BBMRI-NL, a research infrastructure financed by the Dutch government (NWO 184.021.007). Wouter P. te Rijdt is supported by Young Talent Program (CVON PREDICT) grant 2017T001 from the Dutch Heart Foundation. Netherlands Heart Institute, Utrecht, the Netherlands.

## Author contributions

P.D., S.D., J.H., and L.F. wrote the manuscript. J.K., H.B., K.A., C.D., P.Z., E.G., P.A., J.J., C.R., R.S., B.S., W.K., M.S., L.F. edited the manuscript. L.F. conceived the method. P.D., S. D., J.K., L.F. developed the statistical methods. P.D., S.D., J.K., H.B., P.F., T.G., L.F. wrote the software. P.D., J.H., K.A., C.D., P.Z., E.G., K.V., R.K., P.A., S.J., E.H., W.R., Y.V., J.J., C.R., R.S., B.S., W.K., E.Z., J.B. processed, analyzed and interpreted the solved and unsolved cases. J.H., J.K. and H.B. contributed equally to this work.

## Additional information

**Competing interests:** The authors declare no competing interests.

