## [Peer Review File · Nature Communications]

Reviewers' Comments:

Reviewer #1:

Remarks to the Author:

In this manuscript Deelen et al. describe a computational method to prioritize disease candidate genes by looking at the similarity in expression between annotated genes and a set of query genes across a large compendium of human RNAseq data. The manuscript is clearly written. The method proposed here could be of interest to the community if its complementarity or superiority to other existing methods can be clearly shown, of which at this point I'm not fully convinced, but perhaps the authors can clarify some issues:

Major:

- The authors make claims about the novelty of the method yet however do not include many years of gene prioritization literature into their description of their state-of-the-art many of which already include a similar transcriptional similarity scores alongside many other informative data sources (Moreau & Tranchevent, Nature Reviews Genetics volume 13, pages 523–536 (2012)). Instead they compare to a single prioritization method, Exomizer, which has a very particular prioritization method using orthology in animal knockouts. The authors need to clarify what indeed is the novelty of their approach.

- How reasonable is it to estimate performance on exomes which have already been diagnosed? These cases are clearly biased towards having variants in identifiable genes. Only in about 20% of these cases does GADO prioritize the known causative genes above the author's threshold, whereas currently in cohorts of undiagnosed cases the diagnostic rate without using gene prioritization methods lies around ~35-40 (<https://www.ddduk.org/updates.html>).

- Why restrict the analysis to 7 undiagnosed cases? There are publicly available exome sequence datasets of many thousands of undiagnosed cases of Mendelian disorders (e.g. the DDD study). Applying GADO on such datasets would give a much better view of the true performance of the tool than in this limited set of cases.

- If there truly is a problem of false positive gene associations as the authors claim, and GADO relies on known genes to predict transcriptionally similar genes, aren't then GADO predictions sensitive to false positive input data? How do these affect GADO performance?

- It's unclear if the z-scores represented in Figure 3A are computed excluding the gene in question from the HPO-geneset before computing the z-score or not. Most OMIM genes will have been associated with HPO-terms a priori, resulting in an inflation of the z-scores for known gene-disease associations. The same argument holds for the computation of the correlations between GADO z-scores and Clinvar variants/ExAC missense constraint

- The sentence starting at line 268 doesn't make logical sense. How can one prioritize novel disease genes in solved cases? Also what do the authors mean when they say "we do not use the existing annotations"? Don't they compute the z-score based on the transcriptional similarity of genes associated with the HPO-term? If these are known genes are these then not included in the computation of the z-score?

Minor:

- How does propagating information over the HPO tree (when not enough true positives are known) affect the performance of GADO?

- Clustering HPO terms based on GADO predictions can indeed probably identify terms with similar

transcriptional profiles. Can this be due to HPO terms having similar input gene annotations?

- The authors mention that performance in REACTOME is better than in HPO and that one of the causes could be more causative heterogeneity for HPO terms. Can a solution for this be subclustering input gene-disease associations and performing multiple prioritizations per term for each subcluster (and then selecting the maximum score for example)?

Reviewer #2:

Remarks to the Author:

The authors present a method for the analysis of diagnostics exome data that exploits RNA-seq data from a range of tissues and cell types, and using gene co-regulation to predict gene functions and disease annotations. In contrast to existing methods, GADO looks at genes predicted to cause a certain HPO feature based on their involvement in other diseases. To do this, they have assembled a very large dataset (>31,000 samples), used PCA on the correlation matrix to remove low quality samples.

Overall the methodology is new and well presented and the performance reaches interesting levels. The technical details in the paper are described in an occasionally imprecise way. The authors also seem to take for granted that there is a relationship between gene expression and phenotype but in reality the relationship between these two features is extremely complicated and not always reliable (there is an interesting paper that the authors may wish to read on this topic: Feiglin A, Allen BK, Kohane IS, Kong SW. Comprehensive Analysis of Tissue-wide Gene Expression and Phenotype Data Reveals Tissues Affected in Rare Genetic Disorders. Cell Syst. 2017 Aug 23;5(2):140-148.e2).

Major Comments

1. The methods are often very difficult to follow in detail. For instance, the authors should provide more of an explanation of their use of PCA for prediction; I found the explanations a little short on detail. As another instance, the authors conducted a T test based on the eigencoefficients of the genes in a gene set vs background – I do not follow this part of the analysis and the authors should motivate it and describe the concrete analysis steps. Also, there are many ways to generate a combined p-value (z score), and this needs to be described more exactly. One method is mentioned in ref 22, but the authors do not say why they chose this method and whether they tested any others.

The authors state "For each gene-set, we calculated an Area Under the Curve (AUC), using a Mann-Whitney U test..." This is an unusual (but valid) way of doing this, and it is unclear why the authors are emphasizing this. Could they provide a reference?

In many places in the text it is unclear to me whether the prediction scores are for pathways or for diseases/phenotypes. This should be written in a clearer way.

The authors write that if an HPO term cannot be used (because it has too few annotated diseases/genes) GADO will make suggestions for suitable parent terms. It is unclear to me why one would want to do this and there is no prediction problem here because the parent terms can just be looked up?

"a GADO option which ensures any gene associated to a specific HPO term obtains a minimum z-score of 3." => The authors should motivate why this heuristic was necessary. They should define what they mean by a "specific" HPO term (there is no natural definition for this as far as I know).

"If any of the HPO terms did not have significant predictive power, the parent terms were used."

=> I don't understand – shouldn't the more specific (child) terms have more predictive power? And since the terms are taken from the OMIM annotations, they should all have some annotated disease? Or were terms without any annotations taken?

"For 11 of the previously solved cases, GAVIN did not flag the causative variant as a candidate. To be able to include these samples in our GADO benchmark, we added the causative genes missed by GAVIN for these cases manually to the candidate list."

□ This appears to be a very serious limitation of the method presented here and yet the authors do not discuss this as a limitation in the discussion.

"If a patient harbored multiple causative variants in different genes, in case of di-genic inheritance or two inherited conditions, the median rank of these genes was reported"

□ It would be very unusual to have many of these cases in the small cohort (n=83) that was tested here. Can the authors provide more detail about these unusual cases?

2. In any case it is unclear to me why the authors predict gene functions in this manuscript; this aspect is not well connected to other parts of the results, and appears to be used only to validate the methodology. If so, it could be shortened or moved to the supplement.

Is there a difference between GADO Prioritization score and "prediction z score" and "prediction score"? If not, please use one name consistently, and if so please explain.

3. It should be noted that the authors apparently are using all of the features associated with a disease (on average 15 per disease gene) as input to GADO. Many of the clinical tools that use HPO have not been tested with all terms (which often makes it easy/trivial to predict the gene/disease) but with a smaller, randomly chosen set of about 5 terms -- this is done to simulate "real-life situations" in which doctors do not have time to input all of a patient's phenotypes and also because real patients usually do not have all of the phenotypes listed in OMIM/HPO for any given disease. Can the authors comment on this? It seems to me that the validation is technically correct but it is not addressing a real-life challenge.

4. Can the authors provide a few examples of genes with poor prediction scores. Is this a reliable way of predicting a disease-gene correlation (compared to say the ClinGen resource?)

5. The authors do not appear to explicitly compare their method with any previous tools. Can the authors provide a comparison against some of the other phenotype driven exome tools mentioned in the introduction?

Response to reviewers:

We would like to thank both reviewers for their valuable comments. We feel this has helped us to improve our manuscript substantially. We provide point-by-point responses to each of the comments that have been raised:

Reviewer 1:

“In this manuscript Deelen et al. describe a computational method to prioritize disease candidate genes by looking at the similarity in expression between annotated genes and a set of query genes across a large compendium of human RNAseq data. The manuscript is clearly written. The method proposed here could be of interest to the community if its complementarity or superiority to other existing methods can be clearly shown, of which at this point I'm not fully convinced, but perhaps the authors can clarify some issues:

Major:

- The authors make claims about the novelty of the method yet however do not include many years of gene prioritization literature into their description of their state-of-the-art many of which already include a similar transcriptional similarity scores alongside many other informative data sources (Moreau & Tranchevent, Nature Reviews Genetics volume 13, pages 523–536 (2012)). Instead they compare to a single prioritization method, Exomizer, which has a very particular prioritization method using orthology in animal knockouts. The authors need to clarify what indeed is the novelty of their approach.”

The novelty of our approach is that GADO can prioritize genes using only HPO terms, even for genes for which very little to nothing is known. We believe this sets our method apart from existing methods.

We acknowledge that in our initial manuscript we had only compared GADO to Exomizer, but that other methods do exist as well. However, these existing methods have several limitations:

- AMELIE is also able to use multiple HPO terms to prioritize genes and previously has been shown to be very accurate in prioritizing known disease genes. However, it is unable to identify novel disease-gene associates since it is fully dependent on literature. Because identification of novel genes is the primary aim of GADO, we cannot benchmark it against AMELIE.
- The Moreau & Tranchevent review (suggested by the referee) discusses three other gene prioritization tools: Endeavour, ToppGene, and Suspects. Although these tools can be used to predict novel disease gene, they cannot directly use HPO terms, but require sets of genes as input. To overcome this problem, such that these methods can be used, an alternative strategy can be used: Use as input for these methods those genes that

are known to cause any of the HPO terms that apply to a patient. However, if a gene is known to cause more than one of these phenotypes (thus being fairly specific for the symptoms of that specific patient), it will not get a higher weight when prioritizing new genes. GADO does take this explicitly into account, and weighs per gene the evidence for each HPO term. We acknowledge that in principle these methods could be run manually for each separate HPO term, each time taking only the genes, known to cause that particular HPO term. However, methodology to subsequently combine the lists of prioritized genes for each of the HPO terms should then still be devised.

However, in order to get some insight on how well GADO performs in comparison to Endeavour & ToppGene, we have now performed a comparison using our unsolved cases with these methods that show the added value of GADO when prioritizing novel disease genes. These results are presented in the manuscript (“Reanalysis of previously unsolved cases”) and are detailed in supplementary table 6. Regretfully, we were unable to use Suspects: the Suspects website is not functioning anymore, and we were unable to get in touch with the Suspects authors.

“How reasonable is it to estimate performance on exomes which have already been diagnosed? This cases are clearly biased towards having variants in identifiable genes. Only in about 20% of these cases does GADO prioritize the known causative genes above the author’s threshold, whereas currently in cohorts of undiagnosed cases the diagnostic rate without using gene prioritization methods lies around ~35-40 (<https://www.ddduk.org/updates.html>).”

We fully agree with the reviewer, that the benchmarking of gene prioritization methods using only solved cases might lead to an overestimation of the performance, since well studied genes potentially might be easier to prioritize. For instance, if a gene is known to cause a certain phenotype, and this knowledge has been used while training the algorithm, a benchmark that specifically tests how well that gene - phenotype combination ranks will result in strong inflation of the actual performance of that method. GADO does not suffer from this bias: we previously showed that no bias exists towards well studied genes (see supplementary figure 6), but more importantly, when benchmarking how well a known gene - phenotype combination ranks in GADO, we retrain GADO while explicitly excluding knowledge on this gene - phenotype combination.

However, in order to get realistic estimates on the performance of GADO, we have now used a large set of 61 **undiagnosed** cases. These are thus patients that we had already analyzed using our in-house diagnostic exome-sequencing pipeline, which incorporates GAVIN (<https://pediatrics.aappublications.org/content/140/4/e20162854.abstract>), but for whom no causal mutation had been identified yet. We ran GADO on these patients, and followed up on average 2.9 genes per patient that had been assigned a high significance score (Z-score ≥ 5). We then identified 10 patients (16.4%) with a very likely causal gene: these were genes for which we could find additional evidence in literature that supported their role in causing the phenotypes that were observed in these patients.

We used this threshold ($Z\text{-score} \geq 5$) to limit the number of genes that required literature study, but acknowledge that a somewhat more lenient threshold might have resulted in the identification of causative genes for more previously unsolved cases. We believe though that this current threshold provides a reasonable balance between the number of genes to follow-up and the diagnostic yield of those genes that need to be followed-up. This is supported when using the same strategy in 83 **solved** cases: GADO then prioritizes on average 1.1 genes with a $Z\text{-score} \geq 5$ per patient. For 17 cases (20%), the known causative gene then has a $Z\text{-score} \geq 5$, which is quite similar to the proportion of unsolved cases (16.4%) with a very likely causative gene that GADO prioritized.

“Why restrict the analysis to 7 undiagnosed cases? There are publicly available exome sequence datasets of many thousands of undiagnosed cases of Mendelian disorders (e.g. the DDD study). Applying GADO on such datasets would give a much better view of the true performance of the tool than in this limited set of cases.”

We apologize for having caused this confusion: in the original manuscript we had erroneously stated that we had **used** 7 undiagnosed cases. This should have read as that we had **solved** 7 previously undiagnosed cases.

We therefore fully understand that the reviewer requested to use more samples, and we appreciate the suggestion to use the Deciphering Developmental Disorders (DDD) study. However, it is not feasible to study thousands of undiagnosed samples for the following reasons:

- The interpretation of undiagnosed exome sequence datasets is very labor intensive, since for each patient we ascertained the plausibility of the prioritized genes from GADO in multiple ways:
 - We discussed the individual gene and corresponding rare variant with both a clinical geneticist and technician from our department.
 - We ascertained whether these rare variants were de-novo mutations for the patients when trio data was available.
 - We studied literature for evidence this gene or orthologues of this gene in other model organisms can cause phenotypes as observed in our patients.
- Although intellectual disability (ID, HPO term HP:0001249) is the phenotype that applies to each of the DDD samples, this phenotype is caused by a very large set of genes, involved in many different biological processes. This is clearly visible when visualizing these genes while using our co-expression data for inferring relationships between these genes (<https://www.genenetwork.nl/network/HP:0001249>, it may take a minute to load due to the large number of genes): numerous clusters emerge, all having different biological functions (by clicking on the individual clusters it is possible to do pathway analysis per cluster).

In order to get a realistic estimate how well GADO is able to identify causative genes for previously unsolved cases, we have now included all 61 unsolved cases that we have in-house for which exome sequencing and informed consent for research purposes is available. We then identified very likely causative genes for 10 cases (16.4%), as mentioned above.

“If there truly is a problem of false positive gene associations as the authors claim, and GADO relies on known genes to predict transcriptionally similar genes, aren't then GADO predictions sensitive to false positive input data? How do these affect GADO performance?”

We fully agree that false-positive HPO term-gene associations indeed adversely affect the prediction performance of GADO: When false positive genes that have been assigned to an HPO term, it becomes more difficult to establish which gene co-expression principal component are informative for discriminating between the genes that are associated to this HPO terms and all other genes. Although his problem likely applies to most prioritization algorithms that rely upon existing disease-gene associations, we checked systematically how false-positive HPO - gene assignments impact to what extent HPO terms can be accurately predicted for individual genes using the gene co-expression principal components: for each HPO term with an certain number of known gene annotations, we added a set of randomly selected genes, such that the number of known gene annotations increased with 10%. We then redid our gene - HPO term prediction and calculated for each of the HPO terms the area-under-the-curve (AUC), which indicates how well it is possible to predict the genes that cause that HPO term using gene expression data. As expected, we observed a subtle drop of the AUC: the original median AUC was 0.73 and decreased to 0.71, when adding 10% noise. This drop was consistent for each of the HPO terms (Pearson correlation $r = 0.97$, see figure below). As such we conclude that false positive gene associations have a small impact on the performance of GADO, and we have now added these results to the supplement.

“It’s unclear if the z-scores represented in Figure 3A are computed excluding the gene in question from the HPO-geneset before computing the z-score or not. Most OMIM genes will have been associated with HPO-terms a priori, resulting in an inflation of the z-scores for known gene-disease associations. The same argument holds for the computation of the correlations between GADO z-scores and Clinvar variants/ExAC missense constraint”

Reassuringly, we computed the Z-scores represented in figure 3a, while excluding the gene in question from the HPO-geneset before computing the Z-score, such that any prior annotation for that gene will have any effect on the calculated Z-score for that gene. To do this, we used a leave-one-out procedure to calculate the prioritization Z-scores of known associations: when for

a given HPO term a gene already has been assigned to that HPO term, we retrain which principal components are informative for predicting this HPO term while excluding any annotation for that gene, ensuring there is no inflation of Z-scores for known genes. To make this more explicit throughout the manuscript we have now described this in more detail in the methods section.

“The sentence starting at line 268 doesn't make logical sense. How can one prioritize novel disease genes in solved cases? Also what do the authors mean when they say “we do not use the existing annotations”? Don't they compute the z-score based on the transcriptional similarity of genes associated with the HPO-term? If these are known genes are these then not included in the computation of the z-score?”

We apologize for these very unclear formulations. To resolve this, we have now updated this section to better explain that we have used a leave-one-out procedure (as mentioned above): when we make HPO inferences for individual genes that are already associated to a specific HPO term “we do not use the existing annotations”. This ensures there is no inflation of Z-scores for genes with existing HPO annotations, as compared to genes without HPO annotations. A major advantage of this is that we can subsequently conduct an *in-silico* benchmark in solved cases where we can test to what extent GADO is able to prioritize the actual causal gene. We believe such a benchmark is thus also informative for estimating how well GADO is able to correctly predict novel disease genes (see our response above on these results). To improve the readability of our manuscript we have now rephrased the sentences “prioritizing novel disease genes in solved cases” and “we do not use the existing annotations”.

“Minor:

How does propagating information over the HPO tree (when not enough true positives are known) affect the performance of GADO?”

In general we find that using more specific terms yields the most accurate predictions. We therefore only use the parent terms as a last resort since this is better than not using a feature at all. We have now checked how detrimental this is: on average the AUC of a parent term is only 0.04 lower than the AUC of the child term, indicating that using this approach does not have a strong negative impact on the performance of GADO.

“Clustering HPO terms based on GADO predictions can indeed probably identify terms with similar transcriptional profiles. Can this be due to HPO terms having similar input gene annotations?”

Yes, this likely occurs when using HPO terms that share the same genes. However, this will not affect the most interesting application of the clustering of HPO terms as we are interested to identify uncorrelated HPO-terms that are not caused by a shared gene.

“The authors mention that performance in REACTOME is better than in HPO and that one of the causes could be more causative heterogeneity for HPO terms. Can a solution for this be subclustering input gene-disease associations and performing multiple prioritizations per term for each subcluster (and then selecting the maximum score for example)?”

Thank you for this excellent suggestion. We have now conducted some analyses that indeed suggest that by first determining subclusters, and subsequently doing the HPO term prediction for each individual subcluster, performance can improve. However, since we experienced quite a few challenges with respect to overfitting, defining an optimal number of subclusters and the choice of specific clustering algorithm, we believe it is beyond the scope of the paper to implement this. We however have now add a section to the discussion where we indicate how the GADO methodology could be further improved by using subclustering of HPO terms.

Reviewer 2:

“The authors present a method for the analysis of diagnostics exome data that exploits RNA-seq data from a range of tissues and cell types, and using gene co-regulation to predict gene functions and disease annotations. In contrast to existing methods, GADO looks at genes predicted to cause a certain HPO feature based on their involvement in other diseases. To do this, they have assembled a very large dataset (>31,000 samples), used PCA on the correlation matrix to remove low quality samples.

Overall the methodology is new and well presented and the performance reaches interesting levels. The technical details in the paper are described in an occasionally imprecise way. The authors also seem to take for granted that there is a relationship between gene expression and phenotype but in reality the relationship between these two features is extremely complicated and not always reliable (there is an interesting paper that the authors may wish to read on this topic: Feiglin A, Allen BK, Kohane IS, Kong SW. Comprehensive Analysis of Tissue-wide Gene Expression and Phenotype Data Reveals Tissues Affected in Rare Genetic Disorders. Cell Syst. 2017 Aug 23;5(2):140-148.e2).”

We thank the reviewer for these highly valuable comments. We apologize for the sometimes imprecise way of describing technical details throughout the paper. We have now improved upon this substantially (as explained below in our second response to reviewer 2).

We fully agree that the relationship between gene expression and disease phenotypes is complicated: in our original submission we explicitly described a few examples where the use of gene expression is not informative for implicating genes in disease. For instance, we were unable to use gene expression data to infer that the gene TTR causes amyloidosis. Yet, for most other genes, we have observed that gene expression data can be used to correctly infer disease phenotypes. This was also one of the motivating reasons to propose a ‘gene

predictability score', which indicates per gene to what extent gene expression data is actually informative for making functional inferences.

A major differences between our work and the work of Feiglin *et al* is that their predictions are based on tissue-specific gene expression: they prioritize genes that are highly expressed in the tissue relevant to a disease, assuming these are more likely to be related to a disease than genes that show lower expression in that tissue. In contrast, the co-regulation on which GADO is based is only partly driven by genes being specifically expressed in certain tissue but also by having shared expression patterns within a tissue. To make this clear we have now explicitly stated that our co-regulation scores do not solely rely on tissue specificity.

“Major Comments

1. The methods are often very difficult to follow in detail. For instance, the authors should provide more of an explanation of their use of PCA for prediction; I found the explanations a little short on detail. As another instance, the authors conducted a T test based on the eigencoefficients of the genes in a gene set vs background – I do not follow this part of the analysis and the authors should motivate it and describe the concrete analysis steps. Also, there are many ways to generate a combined p-value (z score), and this needs to be described more exactly. One methods is mentioned in ref 22, but the authors do not say why they chose this method and whether they tested any others.

The authors state “For each gene-set, we calculated an Area Under the Curve (AUC), using a Mann-Whitney U test...” This is an unusual (but valid) way of doing this, and it is unclear why the authors are emphasizing this. Could the provide a reference?”

We apologize that our method description was unclear. In short, our GADO method entails the following steps:

1. We first collected a gene expression dataset of 31,499 high-quality RNA-seq samples, reflecting many different tissues and cell-types. We corrected this dataset for various technical confounders.
2. We subsequently performed principal component analysis on the gene correlation matrix and identified 1,588 robustly estimated components.
3. We then study per HPO term each individual principal component, ascertaining whether the eigenvector coefficients of that component are significantly different between the genes that have been explicitly assigned to this HPO-term and the other genes, by conducting a T-test and converting the corresponding T-statistic to a Z-score.
4. As such for each HPO term we now have 1,588 Z-scores, denoting the informativity of each component for predicting that HPO term. We then ascertain to what extent individual genes are likely part of this HPO term: we take per gene the 1,588 eigenvector coefficients, and correlate these to the vector of 1,588 Z-scores (that had been calculated in step 3). This yields a correlation and P-value (indicating the probability that this gene has a role in this HPO term) and we finally convert this P-value into a Z-score.

5. Once we have calculated for each gene – HPO term combination a Z-score, we can study individual patients: for a given patient we take the HPO terms and aim to identify those genes that have a high Z-score for each of these HPO terms. To do this, for each gene we combine the corresponding HPO Z-scores using a Z-transform test = $\text{sum}(\text{Z-scores}) / \text{sqrt}(\text{number of HPO terms})$. This method is sometimes also called ‘Stouffer’s method’. We regret that we previously incorrectly stated that we used an weighted Z-score method to combine these Z-scores.

The reason we have used the Mann-Whitney U-test for determining the AUC is because we also use the U-test to calculate a P-value between prioritization Z-score for genes annotated to a gene-set and the other genes. We only included gene-sets if this difference was significant. We apologize for the confusion that we might have caused when describing the use of a Mann-Whitney U test in order to calculate the AUC: we use a specific library when conducting a Mann-Whitney U test which also yields an AUC. To prevent any confusion we have now do not mention the use of a Mann-Whitney U test anymore.

“In many places in the text it is unclear to me whether the prediction scores are for pathways or for diseases/phenotypes. This should be written in a clearer way.”

The prediction scores throughout the manuscript pertain to disease phenotypes (i.e. HPO terms). We have now clarified upon this in the methods section of our manuscript.

“The authors write that if an HPO term cannot be used (because it has too few annotated diseases/genes) GADO will make suggestions for suitable parent terms. It is unclear to me why one would want to do this and there is no prediction problem here because the parents terms can just be looked up?”

Indeed it is always possible to manually lookup the parent term(s) if a term cannot be used. However, we thought this might be inconvenient to the end user, particularly since it sometimes happens that direct HPO parents are also not usable (see supplementary figure 1 for more details on this), and therefore ‘grandparents’ should be chosen. By providing this functionality, we believe we make the GADO website more user friendly.

“a GADO option which ensures any gene associated to a specific HPO term obtains a minimum z-score of 3.” => The authors should motivate why this heuristic was necessary. They should define what they mean by a “specific” HPO term (there is no natural definition for this as far as I know).”

We agree this heuristic might seem arbitrary, but we believe there is a very valid reason to use it: GADO calculates a Z-score for every HPO-gene combination. This Z-scores denotes the probability that a gene cause a specific HPO term. However, we developed our algorithm in such a way that when a gene has been annotated to a specific HPO term, the Z-score for this combination will be calculated, while ignoring knowledge on this specific HPO-gene

combination. This was intentional, such that HPO predictions are not biased towards genes with known HPO-gene combinations, and therefore these predictions could be used for *in-silico* benchmarks of solved cases: by running GADO on a set of solved patients with HPO terms and exome sequencing available we could subsequently check how well the known causal genes rank.

However, these unbiased predictions can sometimes cause problems when using GADO in clinical practice, because GADO cannot predict every known gene-HPO combination accurately. As such some of these known gene-HPO combinations might have rather insignificant Z-scores. By enforcing that for known gene-HPO combinations, these Z-scores are at least 3, we ensure that known gene-HPO combinations will get prioritized, and that GADO users will not accidentally miss out on known gene-HPO combinations.

*“If any of the HPO terms did not have significant predictive power, the parent terms were used.”
=> I don't understand – shouldn't the more specific (child) terms have more predictive power?
And since the terms are taken from the OMIM annotations, they should all have some annotated disease? Or were terms without any annotations taken?”*

Indeed, specific terms generally have more predictive power, and that is why we start with terms that are as specific as possible. However, there are a few instances when these highly specific HPO terms cannot be used: 1) Some HPO terms exist that have not yet been annotated to any gene. 2) Some HPO terms contain fewer than 10 annotated genes: GADO is unable to make reliable inferences for HPO terms using gene expression data when less than 10 genes have been annotated to that HPO term. 3) For some terms we have over 10 genes annotated, but our gene expression data is not informative and therefore this geneset cannot be accurately predicted. In these instances, GADO uses the more generic parent term in order to have at least some predictive power.

“For 11 of the previously solved cases, GAVIN did not flag the causative variant as a candidate. To be able to include these samples in our GADO benchmark, we added the causative genes missed by GAVIN for these cases manually to the candidate list.”

This appears to be a very serious limitation of the method presented here and yet the authors do not discuss this as a limitation in the discussion.”

We did not discuss this limitation, as this problem relates to GAVIN, which is a previously published method for processing VCF files and classifying variants using ExAC, SnpEff and CADD. Typically, GAVIN yields a list of approximately 200 variants per patient for follow-up analysis, but regrettably had not flagged the causative variant for each of our previously solved cases. However, GADO can also be combined with any other existing variant selection method (that might be better in flagging these causative variants), since GADO only requires HPO terms. We have now stated this limitation clearly in the discussion section.

“If a patient harbored multiple causative variants in different genes, in case of di-genic inheritance or two inherited conditions, the median rank of these genes was reported”

It would be very unusual to have many of these cases in the small cohort (n=83) that was tested here. Can the authors provide more detail about these unusual cases?”

We agree this might seem very unusual, but recently, Posey, J. E. et al. reported (<https://www.nejm.org/doi/full/10.1056/NEJMoa1516767>) that approximately 5% of the patients with a rare Mendelian disease have a second disease. As such, it is quite possible that within (small) cohorts, patients are present that have mutations in multiple genes. Unfortunately, we only had permission to report on HPO terms and causative genes of these cases and were not allowed to discuss these cases in more detail.

“2. In any case it is unclear to me why the authors predict gene functions in this manuscript; this aspect is not well connected to other parts of the results, and appears to be used only to validate the methodology. If so, it could be shortened or moved to the supplement.”

While we use these gene functions to validate our methodology, we also use these predicted gene functions as the basis for our ‘gene predictability score’ metric: this metric is determined per gene by calculating the variance of these predicted gene function Z-scores. As a convenience to other users, we have made these gene function predictions available on the GADO website.

“Is there a difference between GADO Prioritization score and “prediction z score” and “prediction score”? If not, please use one name consistently, and if so please explain.”

“Prediction z score”, “prediction score” and “prioritization Z-score” indeed all refer to the same metric. We have now updated the manuscript and have used prediction Z-score throughout the manuscript.

“3. It should be noted that the authors apparently are using all of the features associated with a disease (on average 15 per disease gene) as input to GADO. Many of the clinical tools that use HPO have not been tested with all terms (which often makes it easy/trivial to predict the gene/disease) but with a smaller, randomly chosen set of about 5 terms -- this is done to simulate “real-life situations” in which doctors do not have time to input all of a patient’s phenotypes and also because real patients usually do not have all of the phenotypes listed in OMIM/HPO for any given disease. Can the authors comment on this? It seems to me that the validation is technically correct but it is not addressing a real-life challenge.”

We fully agree that in a real-life situation doctors will input only a limited number of phenotypes. However, in our benchmark we try to predict back known disease-gene associations, while using all phenotypes that are listed in OMIM. For diseases, for which many phenotypes have been listed, this will likely yield more significant Z-scores than what is likely to be expected in a realistic clinical setting. In order to account for this, we have now performed an analysis that

uses at most 5 HPO-terms (when for a given disease more than 5 HPO-terms are known, 5 of these HPO terms are randomly selected). We then observe a correlation of 0.86 with prioritization Z-score when using all HPO-terms (see figure below, data points on the diagonal reflect diseases with 5 or fewer HPO terms).

This indicates that GADO also works well when using only 5 HPO terms, but we believe this is an underestimate, since we randomly select 5 of the annotated HPO terms per disease. We expect that in reality, clinicians will try to enter HPO terms that describe clearly different phenotypes, yielding more informative results.

“4. Can the authors provide a few examples of genes with poor prediction scores. Is this a reliable way of predicting a disease-gene correlation (compared to say the ClinGen resource?)”

Resources such as ClinGen are extremely useful, since they adhere to very strict guidelines, and therefore have very reliable entries. GADO by no means aims to replace such efforts. If

there was sufficient evidence to annotate a gene-disease association as “definitive” in ClinGen, we would trust that conclusion, regardless of the GADO prediction score for that gene - phenotype combination, because GADO sometimes does not work. Listed below are the ten disease-gene combinations with the lowest GADO prioritization Z-scores (but with a gene predictability score above 1, which indicates that gene expression data should actually be informative for making functional inferences):

- OMIM:615217 (Ataxia-oculomotor apraxia 3) - *PIK3R5*
- OMIM:603383 (Glaucoma 1, open angle, F) - *ASB10*
- OMIM:221770 (Polycystic lipomembranous osteodysplasia with sclerosing leukoencephalopathy 1) - *TYROBP*
- OMIM:190685 (Down syndrome) - *GATA1*
- OMIM:605726 (Spinal muscular atrophy, distal, autosomal recessive, 2) - *RAX2*
- OMIM:260920 (Hyper-IgD syndrome) - *MVK*
- OMIM:217090 (Dysplasminogenemia) - *PLG*
- OMIM:300971 (Bartter syndrome, type 5, antenatal, transient) - *MAGED2*
- OMIM:616039 (Charcot-Marie-Tooth disease, recessive intermediate D) - *COX6A1*
- OMIM:119500 (Popliteal pterygium syndrome 1) - *IRF6*

It is evident from this list that gene expression data (and thus GADO) is not always informative: for instance *PLG* (plasminogen) is not predicted for the disorder dysplasminogenemia. However, some other disease-gene combinations with low GADO Z-scores actually are false-positives: for instance, although *GATA1* maps to chromosome X, it is currently annotated as causing Down syndrome, whereas it should not. As such, GADO might also be of use to identify false-positive annotations in literature, although we do realize that disproving previously reported associations is very difficult. We have now added a supplementary table with the prioritization Z-scores and predictability scores for all OMIM disease-gene associations.

“5. The authors do not appear to explicitly compare their method with any previous tools. Can the authors provide a comparison against some of the other phenotype driven exome tools mentioned in the introduction?”

We previously had only compared the performance of GADO to Exomiser for a set of solved exome-sequenced cases with a known disease genes (figure 4 and supplementary table 4).

We have now also compared GADO to Exomiser, Endeavour and ToppGene for a set of 61 unsolved cases for whom exome-sequencing data was available. We observed that GADO more effectively prioritizes the likely causal genes. We have now added these results to the manuscript and have described detailed statistics in supplementary table 6.

Reviewers' Comments:

Reviewer #1:

Remarks to the Author:

The authors have adequately addressed my previously raised questions concerning the scientific validity and novelty of the proposed method. I however have one last remaining concern:

In the Editorial Policy Checklist the authors have checked that all code and data would be made available as per the policy guidelines mentioned here:

<https://www.nature.com/authors/policies/availability.html>

However, as far as I could see, they only provide access to usage of their tool on their own website, which doesn't offer direct access to the data used by the algorithm, the data used to benchmark the tool nor the code used. If their website would fail to be maintained in the future (as is often the case with academic bioinformatics tools) then reproduction of the results or usage of the tool would be rendered impossible.

- I understand that sharing clinical genomic data can be difficult, but this is usually resolved by uploading the data to public repositories (e.g. EGA) under Data Access Committee approval. In case, due to ethical concerns, the VCF files can not be made available, at least the list per patient of genes carrying potential disease-causing mutations (which should be completely anonymous) should be made available in order to be able to reproduce the proposed benchmark.

- Unless for technical reasons, which should be specified by the authors, it is impossible for users to run this analysis on their own compute infrastructure, the underlying code for the tool should be made available.

- At the very least the computed correlation matrix and a dump of the HPO-term/gene matrix (which were used to compute the results in this manuscript) should be made available in order to let the community reproduce and possibly improve upon the results presented here.

Reviewer #2:

Remarks to the Author:

The authors have addressed my concerns. This is a nice paper! As a final suggestion, one of the authors responses to my my comments about digenic inheritance is incorrect.

Digenic inheritance does not refer to a case where one individual has two different Mendelian diseases (this was what the Posey paper was investigating). Instead, the term digenic inheritance describes a situation in which the clinical expression depends upon the presence of two mutations in two different genes. PMID: 23785127 provides an excellent review of this concept.

Response to reviewers:

We would like to thank the reviewers for their second assessment. Please find below our responses to the final remarks.

Reviewer 1 (Remarks to the Author):

“The authors have adequately addressed my previously raised questions concerning the scientific validity and novelty of the proposed method. I however have one last remaining concern:

In the Editorial Policy Checklist the authors have checked that all code and data would be made available as per the policy guidelines mentioned here:

<https://www.nature.com/authors/policies/availability.html>

However, as far as I could see, they only provide access to usage of their tool on their own website, which doesn't offer direct access to the data used by the algorithm, the data used to benchmark the tool nor the code used. If their website would fail to be maintained in the future (as is often the case with academic bioinformatics tools) then reproduction of the results or usage of the tool would be rendered impossible.

- I understand that sharing clinical genomic data can be difficult, but this is usually resolved by uploading the data to public repositories (e.g. EGA) under Data Access Committee approval. In case, due to ethical concerns, the VCF files can not be made available, at least the list per patient of genes carrying potential disease-causing mutations (which should be completely anonymous) should be made available in order to be able to reproduce the proposed benchmark.”

We are not allowed to share the VCF files of our solved and unsolved cases for legal and ethical reasons. We have now included a supplementary data 3 with genes that are used for benchmarking to ensure our results are reproducible.

“- Unless for technical reasons, which should be specified by the authors, it is impossible for users to run this analysis on their own compute infrastructure, the underlying code for the tool should be made available.”

We have made an open source command line version of the prioritization that is also available on github and the needed data is available on figshare. This allows offline prioritization and ensures reproducibility should our server not be available. The code used to create the prioritization matrix is also available. The binaries, sources, manual, and data files are listed here: <https://github.com/molgenis/systemsgenetics/wiki/GADO-Command-line>.

“- At the very least the computed correlation matrix and a dump of the HPO-term/gene matrix (which were used to compute the results in this manuscript) should be made available in order to let the community reproduce and possibly improve upon the results presented here.”

We have also made all the matrices needed to reproduce our analysis available on figshare. We welcome any improvements to our matrix.

Reviewer #2 (Remarks to the Author):

“The authors have addressed my concerns. This is a nice paper! As a final suggestion, one of the authors responses to my my comments about digenic inheritance is incorrect.

Digenic inheritance does not refer to a case where one individual has two different Mendelian diseases (this was what the Posey paper was investigating). Instead, the term digenic inheritance describes a situation in which the clinical expression depends upon the presence of two mutations in two different genes. PMID: 23785127 provides an excellent review of this concept.”

Thank you for pointing out this error, we indeed used this term incorrectly and have corrected this in the manuscript.